# GATTA:
# Graph Active Learning with Test-Time Augmentation

## Abstract

Test-time augmentation (TTA) has proven effective for improving model robustness and uncertainty estimation in computer vision, yet its application to graph-structured data remains largely unexplored. We introduce GATTA (Graph Active Learning with Test-Time Augmentation), a framework for enhancing active learning by aggregating predictions across multiple augmented views to produce more reliable uncertainty estimates. To address the challenge of label-preserving graph augmentations, GATTA incorporates a consistency-based filtering mechanism that discards augmented views yielding unreliable predictions.

We systematically evaluate GATTA across multiple graph datasets, GNN architectures, and acquisition strategies. Our results show that simple uncertainty-based methods, such as Entropy and Least Confidence, benefit most from TTA, achieving performance competitive with more sophisticated and computationally expensive approaches. GATTA generalizes across architectures, outperforms model-side ensemble methods such as MC Dropout. We further show that GATTA scales efficiently with both ensemble size and graph size. Extensive analysis of augmentation types, strengths, and filtering strategies provides practical guidelines for effective deployment.

Our findings demonstrate that augmenting simple methods with TTA offers a more efficient path to strong active learning performance than engineering complex acquisition functions, enabling practitioners to achieve competitive results with lower computational overhead and reduced implementation complexity.

## 1 Introduction

Graph neural networks (GNNs) have become the dominant paradigm for modeling relational data, achieving state-of-the-art performance across a wide range of applications, from molecular property prediction to social network analysis (Zhou et al., 2020; Wu et al., 2019b). However, most high-performance models still rely on substantial labeled data for tasks like node classification, creating a significant labeling bottleneck. This challenge is particularly acute in scientific and industrial domains where annotation requires costly domain expertise or experimental validation. (Gal et al., 2017; Litjens et al., 2017; Gilmer et al., 2017; Wu et al., 2017; Halbouni et al., 2022)

Active learning (AL) addresses this bottleneck by strategically selecting the most informative nodes for labeling, typically using uncertainty-based acquisition strategies such as Least Confidence (LC), Entropy, or Bayesian Active Learning by Disagreement (BALD). However, graph-structured data presents unique challenges that distinguish it from traditional active learning settings (Hu et al., 2020). The structural dependencies and non-i.i.d. nature of graphs make uncertainty estimation particularly difficult, as node predictions depend on multi-hop neighborhoods rather than being independent samples (Fuchsgruber et al., 2024; Wang et al., 2024). Distinguishing epistemic uncertainty from aleatoric uncertainty becomes further complicated by neighborhood aggregation effects, often leading to biased node selection and suboptimal labeling strategies (Fuchsgruber et al., 2024).

Test-time augmentation (TTA) has proven highly effective in computer vision for improving uncertainty estimation by generating multiple perturbed views of inputs during inference and aggregating their predic-

tions. While TTA has demonstrated clear benefits for image classification (Gaillochet et al., 2022; Wang et al., 2018; Conde et al., 2023), its application to graph-structured data remains largely unexplored (Ju et al., 2023; Bo et al., 2021). This gap presents a significant opportunity: graph-specific augmentations could exploit relational structure to yield more reliable uncertainty estimates for active learning, potentially addressing the fundamental challenges of uncertainty quantification in graph settings.

We introduce GATTA (Graph Active Learning with Test-Time Augmentation), a framework that systematically integrates test-time augmentation into graph active learning pipelines. GATTA combines graph-specific augmentations with two aggregation strategies, GATTA-S (Score Aggregation) and GATTA-P (Prediction Aggregation), and incorporates a consistency-based filtering mechanism that preserves label-relevant properties while discarding potentially misleading perturbations. In this work, we do not consider graph-level classification tasks, but focus on graph active learning for transductive node classification, where labels are iteratively acquired for nodes within a single graph per dataset to improve node classification performance. Through comprehensive evaluation across multiple datasets, GNN architectures, and AL acquisition strategies, we demonstrate that TTA particularly benefits uncertainty-based methods, enabling simple strategies to achieve competitive performance with complex approaches while reducing computational overhead. Our findings suggest a practical design principle: rather than engineering sophisticated acquisition functions, practitioners can augment efficient uncertainty-based methods with TTA to achieve strong active learning performance at lower computational cost.

**Contributions**

1. We introduce GATTA, a framework for test-time augmentation in graph active learning with consistency-based filtering for non-label-invariant augmentations.
2. We demonstrate through comprehensive experiments that simple uncertainty methods with GATTA match complex acquisition functions at lower computational cost.
3. We provide practical deployment guidelines covering augmentation type, strength, and ensemble size.

## 2 Related Work

### 2.1 Active Learning on Graphs

Active learning on graphs presents fundamental challenges that distinguish it from classical active learning paradigms. The structural interdependencies inherent in graph data violate the independence assumption underlying traditional uncertainty sampling methods, as node predictions are influenced by their multi-hop neighborhoods rather than being isolated samples (Kipf & Welling, 2016). This interconnected nature complicates uncertainty estimation, where distinguishing epistemic uncertainty (reducible through additional labels) from aleatoric uncertainty (irreducible data noise) becomes particularly challenging due to information propagation effects (Wang et al., 2024).

Early graph-specific active learning approaches focused on adapting classical strategies to structural settings. Cai et al. (2017) introduced Active Graph Embedding (AGE), which combined graph embeddings with uncertainty and centrality measures to identify informative nodes. Gao et al. (2018) formulated node selection as a multi-armed bandit problem in their ANRMAB framework, while Regol et al. (2020) proposed Graph Expected Error Minimization (GEEM), directly targeting nodes that maximize expected error reduction across the graph structure.

Recent advances have pursued more sophisticated uncertainty quantification strategies. Kang et al. (2022) proposed JuryGCN, a frequentist-based approach that quantifies uncertainty in GCNs using jackknife estimators. Most relevantly, Fuchsgruber et al. (2024) demonstrated that epistemic uncertainty sampling is theoretically optimal for graph active learning, developing practical approximation methods (Multiple Pseudo-Labels (MP) and Expected Single Pseudo-Label (ESP)) that achieve strong empirical performance.

Despite their promise, these sophisticated methods remain computationally demanding, which limits their scalability and ease of adoption. This raises the question of whether simpler, lightweight strategies, aug-

mented with techniques such as test-time augmentation, can achieve similar or even superior performance at a fraction of the cost.

## 2.2 Uncertainty Quantification in Graph Neural Networks

Robust uncertainty quantification in GNNs requires addressing the unique challenges posed by relational data structures. Bayesian approaches have shown promise, with Zhang et al. (2018) developing variational inference methods for GNNs and Lakshminarayanan et al. (2016) proposing ensemble-based uncertainty estimation that accounts for graph structure.

Monte Carlo Dropout has been adapted for graph settings by Gal & Ghahramani (2015), who demonstrated improved calibration on node classification tasks. Meanwhile, Zhuang et al. (2024) explored temperature scaling specifically designed for graph neural networks.

These approaches, although effective, often require architectural modifications or additional training procedures, limiting their applicability to existing models.

## 2.3 Test-Time Augmentation

Test-time augmentation enhances model predictions by generating multiple transformed views of the input during inference and aggregating their outputs. For a model $f$ with parameters $\theta$ and input $x$, TTA computes predictions as:

$$\hat{Y} = \frac{1}{N} \sum_{i=1}^{N} f_\theta(\varphi_i(x)) \tag{1}$$

where $\varphi_i$ represents different augmentation functions. This ensemble approach provides richer uncertainty estimates by capturing prediction variance across multiple input perturbations, particularly valuable for distinguishing epistemic from aleatoric uncertainty (Wang et al., 2018).

TTA has demonstrated consistent improvements across diverse domains. In computer vision, Shanmugam et al. (2021) showed significant gains in medical image segmentation, while Conde et al. (2023) established connections between TTA and model calibration theory. Natural language processing applications have emerged more recently, with Lu et al. (2022) applying TTA to text classification with word- and character-based augmentations.

Recent work has also explored TTA for active learning. Gaillochet et al. (2022) demonstrated that aggregating predictions across augmented image views improves uncertainty estimation for medical image annotation, establishing TTA as a promising mechanism for enhancing active learning acquisition functions. However, whether these ideas transfer directly to graph neural networks remains an open question. Unlike images, graph predictions depend jointly on node attributes and relational structure, making graph perturbations fundamentally different from conventional image augmentations and introducing additional challenges for uncertainty estimation.

## 2.4 Graph Augmentation Techniques

Graph augmentation strategies form the foundation for effective TTA in graph settings. Structural augmentations include edge dropout Rong et al. (2019), where edges are randomly removed during training or inference, and graph subsampling that aims to find augmented graph instances from the input graphs that best preserve desired properties by keeping a portion of nodes and their underlying linkages (Qiu et al., 2020). Node-level augmentations, such as feature masking, Gaussian noise injection, and feature shuffling, target different aspects of graph structure preservation (Ding et al., 2022).

The critical challenge in graph augmentation is maintaining label-relevant information while introducing meaningful diversity. Unlike image transformations such as rotation or cropping, which are generally designed to preserve semantic class labels, graph augmentations directly modify node features and local connectivity, both of which alter the receptive field of a graph neural network. Consequently, even small perturbations may alter the semantic evidence available for prediction. Recent work has therefore focused on designing

augmentations that explicitly preserve semantic consistency. Yue et al. (2022) proposed perturbing graphs in representation space under label-preserving constraints, while Luo et al. (2022) employed reinforcement learning to search for augmentation policies that maintain label invariance. Our propsed filtering ensures that uncertainty reflects semantic ambiguity, not structural instability.

### 2.5 Novelty and Positioning

While graph augmentation techniques (Liu et al., 2021; Zhao et al., 2020) have been explored for computational efficiency (Cui et al., 2022), representation learning during training (Katsimpras & Paliouras, 2024), and class-balancing via reinforcement learning (Yu et al., 2024), GATTA fundamentally differs in both objective and design. Prior work applies augmentation either as a preprocessing step to scale training (Cui et al., 2022), to improve self-training performance (Katsimpras & Paliouras, 2024), or to address class imbalance (Yu et al., 2024). Test-time augmentation (TTA) itself has been used in graph learning but never for uncertainty-driven active learning. Bo et al. (2021) applied TTA for social influence prediction and Ju et al. (2023) used virtual node augmentation to address degree bias.

GATTA is the first framework to systematically integrate test-time augmentation into graph active learning. Unlike prior TTA approaches, which typically assume label-preserving augmentations, GATTA explicitly addresses the challenge of non-label-invariant graph perturbations through a graph-specific consistency mechanism. Beyond introducing this framework, we systematically characterize how test-time augmentation should be deployed in graph active learning by studying its interaction with multiple acquisition strategies, aggregation mechanisms, augmentation choices, and computational trade-offs. This analysis results in practical deployment guidelines, and demonstrates that simple uncertainty-based acquisition strategies can often match or outperform considerably more sophisticated methods when equipped with more reliable uncertainty estimates.

## 3 Method

### 3.1 Acquisition Strategies

In active learning, an *acquisition function* (or query strategy) assigns each unlabeled node a score, indicating its informativeness for model improvement. Given predicted class probabilities $P \in \mathbb{R}^{|V| \times C}$, an acquisition function is a mapping $Q : \mathbb{R}^{|V| \times C} \to \mathbb{R}^{|V|}$, where $Q(P)_v$ denotes the informativeness score for node $v \in V$.

We categorize acquisition strategies into three groups: **simple uncertainty-based methods** (Least Confidence (Wang & Shang, 2014), Entropy (Shannon, 1948)), **complex uncertainty-based methods** (MP, ESP (Fuchsgruber et al., 2024)), and **other methods** (AGE (Cai et al., 2017), ANRMAB (Gao et al., 2018), GEEM (Regol et al., 2020)).

### 3.2 Graph Active Learning with Test-Time Augmentation

We designed GATTA as a plug-and-play module to enhance existing graph active learning strategies without requiring architectural changes. We consider the standard graph active learning problem for transductive node classification, where the objective is to iteratively acquire labels for unlabeled nodes in order to improve node classification performance.

Given an attributed graph $G = (A, X)$, let $q(\varphi \mid G)$ denote the distribution induced by a stochastic graph augmentation process. Rather than evaluating an uncertainty-based acquisition function $Q$ on a single graph realization, GATTA estimates its expectation over the augmentation distribution,

$$Q^* = \mathbb{E}_{\varphi \sim q(\varphi|G)} \left[ Q(f_\theta(\varphi(G))) \right], \tag{2}$$

where $f_\theta$ is a pre-trained GNN. This formulation favors nodes whose uncertainty persists under plausible local graph perturbations rather than relying on a single graph realization.

Since the expectation above is generally intractable, GATTA approximates it using Monte Carlo sampling. Specifically, we sample $N$ augmented graph views

$$G^{(i)} = \varphi_i(G) = (A^{(i)}, X^{(i)}), \quad i = 1, \dots, N, \tag{3}$$

together with the original graph $G^{(0)} = G$, and evaluate the GNN on each view,

$$\mathbf{P}_i = f_\theta(G^{(i)}) \in \mathbb{R}^{|V| \times C}, \tag{4}$$

where $|V|$ is the number of nodes, $D$ is the feature dimension, and $C$ is the number of classes. The resulting predictions $\{\mathbf{P}_i\}_{i=0}^N$ provide Monte Carlo samples used to approximate $Q^*$.

GATTA consists of two key components: graph augmentation, which generates perturbed graph views, and aggregation, which estimates uncertainty-based acquisition scores from these views. The following subsections describe the proposed aggregation strategies (GATTA-P and GATTA-S), the graph augmentation operators, and the consistency filtering mechanism used to improve robustness to non-label-preserving augmentations.

### 3.3 Aggregation Methods: GATTA-S and GATTA-P

The Monte Carlo samples introduced in the previous subsection can be aggregated in multiple ways depending on whether the acquisition function is evaluated before or after prediction aggregation.

**GATTA-S** (Score Aggregation) directly estimates the expected acquisition score by applying the acquisition function $Q : \mathbb{R}^{|V| \times C} \to \mathbb{R}^{|V|}$ independently to each augmented view and averaging the resulting scores:

$$\mathbf{Q}_\mathrm{S} = \mathbb{E}_{\varphi \sim q(\varphi|G)} \left[ Q(\mathbf{P}_\varphi) \right] \approx \frac{1}{N+1} \sum_{i=0}^N Q(\mathbf{P}_i), \tag{5}$$

where $\mathbf{P}_\varphi = f_\theta(\varphi(G))$. This corresponds to a Monte Carlo estimate of the expected acquisition score over the augmentation distribution.

**GATTA-P** (Prediction Aggregation) instead averages the predictive distributions before applying the acquisition function:

$$\mathbf{Q}_\mathrm{P} = Q\left( \mathbb{E}_{\varphi \sim q(\varphi|G)}[\mathbf{P}_\varphi] \right) \approx Q\left( \frac{1}{N+1} \sum_{i=0}^N \mathbf{P}_i \right). \tag{6}$$

The key distinction is that GATTA-S estimates the expected acquisition score directly, whereas GATTA-P computes the acquisition score of the averaged predictive distribution. Although computationally more efficient, as it only requires single evaluation of the acquisition function, GATTA-P is also more sensitive to disagreement among augmented views particularly when graph augmentations are not perfectly label-preserving due to the introduced structural instability. This observation motivates the consistency filtering mechanism introduced in Section 3.5. For entropy-based acquisition functions, an information-theoretic interpretation of this distinction is provided in Appendix B.

### 3.4 Graph Augmentations

We employ three standard graph augmentation strategies: feature masking (You et al., 2020), feature noising (Zhang et al., 2022), and edge dropout (Rong et al., 2019). Feature masking randomly sets node features to zero with probability $p_\mathrm{mask}$, testing model reliance on specific features. Feature noising adds Gaussian noise $\mathcal{N}(0, \sigma_\mathrm{noise}^2)$ to node features, introducing controlled uncertainty while preserving feature magnitudes. Edge dropout randomly removes edges with probability $p_\mathrm{drop}$, testing structural dependencies. These complementary augmentations expose different sources of model uncertainty while aiming to preserve label-relevant information.

### 3.5 Consistency Filtering

Graph augmentations are intended to introduce local perturbations while preserving node semantics. In practice, however, stochastic graph augmentations are not guaranteed to be label-preserving, causing disagreement that may reflect augmentation-induced semantic drift rather than informative uncertainty.

For a node $v$ and augmented view $i$, let $\hat{y}_i^{(v)} = \arg\max_c (\mathbf{P}_i)_{v,c}$ denote the predicted class, and define the consistency event

$$m_i^{(v)} = \mathbb{1}\{\hat{y}_i^{(v)} = \hat{y}_0^{(v)}\},$$

where $\mathbb{1}\{\cdot\}$ is the indicator function and $\mathbf{m}_i \in \{0,1\}^{|V|}$ denotes the vector of node-wise consistency masks. To mitigate semantic drift, we restrict uncertainty estimation to model-consistent perturbations by approximating the conditional expectation $\mathbb{E}[Q(f_\theta(\varphi(G))) \mid \mathbf{m}]$, where prediction consistency with the original graph serves as a practical proxy for local semantic consistency.

The conditional expectation is approximated by Monte Carlo averaging over the subset of consistent augmented views. The filtered acquisition scores for GATTA-S and GATTA-P are therefore computed as

$$\frac{1}{\mathbf{z}} \sum_{i=0}^N \mathbf{m}_i \odot Q(\mathbf{P}_i) \qquad \text{and} \qquad Q\left(\frac{1}{\mathbf{z}} \sum_{i=0}^N (\mathbf{m}_i \mathbf{1}^T) \odot \mathbf{P}_i\right),$$

where $\odot$ denotes element-wise multiplication, $\mathbf{z} = \sum_{i=0}^N \mathbf{m}_i$ counts the number of consistent views for each node, and $\mathbf{1} \in \{1\}^C$ broadcasts the node-wise mask across the class dimension.

Consistency filtering can therefore be interpreted as approximating a conditional expectation over model-consistent perturbations. A confidence-weighted variant is presented in Appendix C, while Appendix B provides an information-theoretic interpretation explaining why prediction aggregation (GATTA-P) is more sensitive to non-label-preserving perturbations than score aggregation (GATTA-S).

## 4 Active Learning Protocol

We consider the *transductive node classification* setting, where the objective is to construct an initial labeled node set $\mathcal{L}$ and iteratively expand it by selecting informative nodes from the unlabeled set $\mathcal{U}$.

We follow the standard graph active learning protocol. Initially, one node per class is randomly selected to form the labeled set. A GNN is trained from scratch on the current labeled set, after which the chosen acquisition strategy scores all unlabeled nodes. The highest-scoring node is then annotated and added to the labeled set. Following Fuchsgruber et al. (2024), the classifier is retrained from scratch after each acquisition round to ensure that performance improvements arise from the acquired labels rather than warm-start optimization. This retraining cycle is repeated until the acquisition budget is exhausted, after which the final model is evaluated on the held-out test set.

Unless stated otherwise, we acquire one node per iteration using a budget of $4C$, where $C$ denotes the number of classes. This matches the protocol adopted in prior graph active learning work, enabling direct comparison with existing methods. Consequently, the final labeled set remains substantially smaller than the training splits typically used in standard semi-supervised node classification benchmarks.

All reported results are averaged over 25 independent trials (5 random initial labeled pools $\times$ 5 random model initializations). To ensure a fair comparison, all model-training hyperparameters are kept identical between each baseline acquisition strategy and its GATTA-enhanced counterpart; GATTA modifies only the inference-time acquisition scores.

## 5 Sensitivity and Configuration Analysis

While hyperparameter optimization is typically constrained in active learning settings due to the limited availability of labeled data, understanding GATTA's sensitivity to different configurations is crucial for practical deployment. We therefore conducted a systematic study of augmentation types, filtering mechanisms,

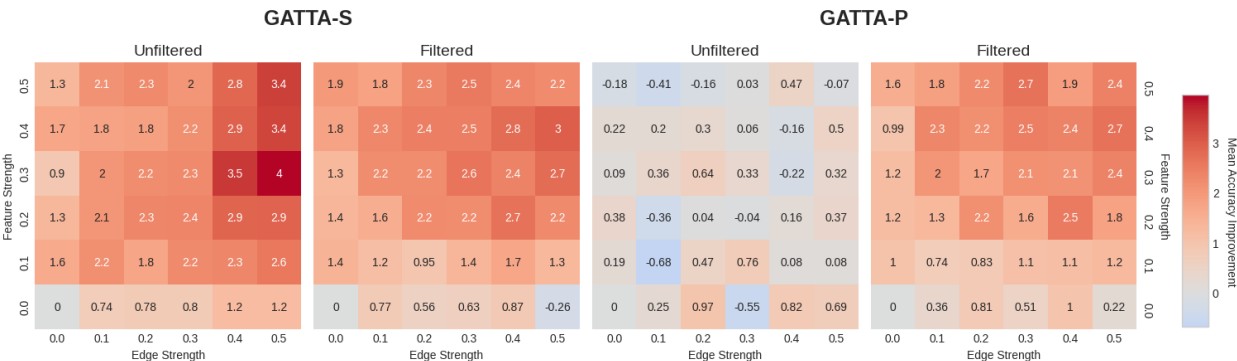

Figure 1: Effect of augmentation strength on performance for FN+ED. Heatmaps show accuracy gains (%) across noise variance ($\sigma^2_{noise} \in [0, 0.5]$) and dropout probability ($p_{drop} \in [0, 0.5]$). Results are reported for GATTA-S (left) and GATTA-P (right), each with and without filtering. Improvements concentrate at higher strengths ($\sim 0.3 - 0.5$). GATTA-S benefits without filtering but filtering broadens the region of effective strengths, while GATTA-P requires filtering to achieve any improvement.

strength parameters, ensemble size, and runtime. Experiments in this section utilize two representative datasets (CoraML Getoor et al. (2005) and PubMed Namata et al. (2012)), two models (GCN Kipf & Welling (2016) and SGC Wu et al. (2019a)), and uncertainty-based strategies (Entropy and Least Confidence). We present our findings denoted with **F**.

Table 1: Comparison of augmentation types and combinations. Reported are the average and 75 percentile performance gains (%) across datasets, models, and augmentation strengths. Combined augmentations, particularly FN+ED, consistently outperform single augmentations.

| Strategy | Average | 75th percentile |
|---|---|---|
| Feature Masking | $0.03 \pm 0.83$ | 0.62 |
| Feature Noising | $\mathbf{0.69 \pm 0.67}$ | **1.21** |
| Edge Dropout | $0.53 \pm 0.84$ | 1.03 |
| FM + ED | $0.53 \pm 0.96$ | 1.16 |
| FN + ED | $\mathbf{1.12 \pm 1.05}$ | **1.94** |

### 5.1 Augmentation Type Selection

We evaluated Feature Masking (FM), Feature Noising (FN), Edge Dropout (ED), and their pairwise combinations across the mentioned models, datasets, and strategies. Table 1 summarizes averaged, augmentation-wise performance.

**F1. Feature Noising outperforms Feature Masking** ($0.69 \pm 0.67\%$ vs. $0.03 \pm 0.83\%$). Additive noise preserves feature scale during neighborhood aggregation, whereas masking creates information voids that propagate through message-passing layers, making FN more suitable for test-time perturbations.

**F2. Combined augmentations consistently outperform single augmentations.** The FN+ED combination achieved the largest average improvement ($+1.12 \pm 1.05\%$). FM+ED showed modest gains ($+0.53 \pm 0.96\%$), comparable to single augmentations. Multi-modal perturbations produce more informative uncertainty signals when both augmentation types contribute meaningfully to the ensemble.

Based on these findings, we adopt FN+ED for all subsequent experiments.

## 5.2 Augmentation Strength and Filtering

We systematically evaluated Feature Noising variance $\sigma_{noise}^2 \in [0.0, 0.5]$ and Edge Dropout probability $p_{drop} \in [0.0, 0.5]$, testing all combinations with and without filtering for both GATTA variants. Figure 1 visualizes the interaction between augmentation strength, filtering, and aggregation strategy.

**F3. Optimal performance occurs at higher augmentation strengths.** With filtering applied, performance peaked at $\sigma_{noise}^2 \in [0.3, 0.5]$ and $p_{drop} \in [0.3, 0.5]$, achieving improvements up to 3.0 over baseline (Figure 1). Stronger perturbations expose more informative uncertainty signals, provided label-inconsistent augmentations are filtered.for For more details, see Appendix D. For comparison with confidence weighted filtering variant, see Appendix C.

**F4. GATTA-S is more robust to augmentation strength and consistency filtering** GATTA-P exhibits strong sensitivity to filtering: without it, the method shows negligible improvement at weak augmentation strengths and active degradation at strong strengths (Figure 1, up to $-0.5\%$ decline), failing overall ($0.12 \pm 1.04\%$) but achieving substantial improvements with filtering ($1.96 \pm 1.27\%$). Prediction-level aggregation directly averages class probabilities across views, so label-inconsistent augmentations corrupt the uncertainty estimate, making filtering essential. In contrast, GATTA-S demonstrates robustness across the entire configuration space, performing best without filtering ($2.49 \pm 1.42\%$) and maintaining stable performance even at strong augmentation levels. Score-level aggregation computes acquisition scores independently per view before averaging, allowing the ensemble to balance label-inconsistent signals without explicit filtering. Across all configurations, GATTA-S achieves greater improvements both on average and at peak performance.

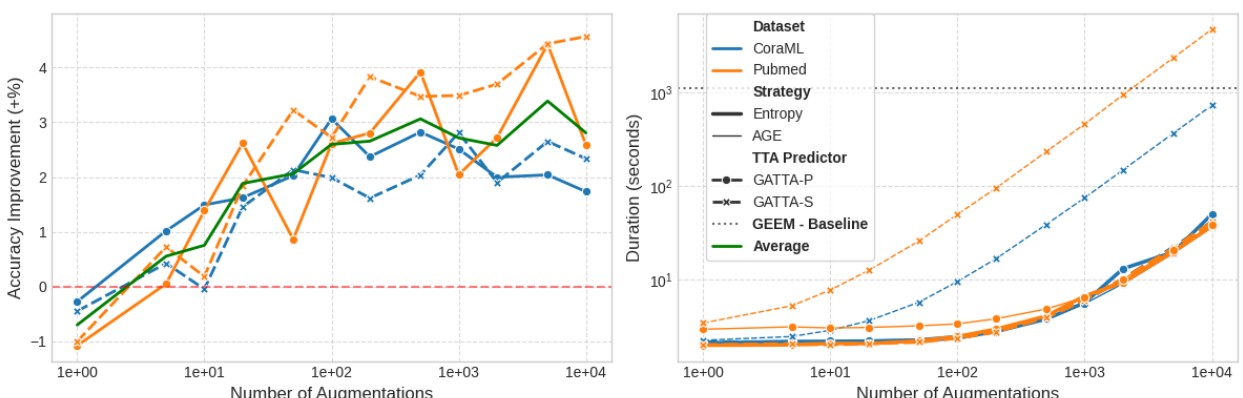

Figure 2: Accuracy improvements (**Left**) and runtime scaling (**Right**) with increasing ensemble size. Accuracy gains saturate around $N \approx 200$. GATTA-S runtime increases drastically for complex strategies (AGE), while both GATTA variants scale well for simple strategies (Entropy)

## 5.3 Ensemble Size, Runtime, and Scalability Analysis

We evaluated ensemble sizes $N \in [1, 10\,000]$ to characterize the performance-cost trade-off. Figure 2 shows accuracy and runtime scaling patterns.

**F5. Performance scales logarithmically with ensemble size.** Accuracy improvements follow Performance $\approx$ baseline$+0.37 \times \log(N+1)$ (Figure 2, right). Substantial gains occur up to $N \approx 200$ ($+2.65 \pm 0.93\%$), with diminishing returns thereafter. Beyond $N = 500$, each additional 100 views yields less than 0.1% improvement, suggesting that moderate ensemble sizes capture most of the uncertainty signal diversity.

**F6. GATTA runtime scales efficiently.** For simple acquisition functions like Entropy, where inference dominates, GATTA-P and GATTA-S scale similarly, both requiring $\sim 2\times$ baseline time at N $=$ 500 (Figure 2, right). For expensive acquisition functions like AGE, where acquisition computation dominates, GATTA-S scales poorly (requiring N+1 AGE evaluations), whereas GATTA-P remains efficient. Crucially,

Table 2: Detailed performance evaluation of GATTA configurations. The table presents average accuracy improvements for GATTA-S (Entropy-S, LC-S) and GATTA-P across diverse datasets, GNN models, and acquisition strategies, highlighting key performance patterns. Values represent average accuracy improvement over baseline methods. MP, ESP, and GEEM were only tested with SGC, as in their original implementation. Additionally, ESP and GEEM were not evaluated on the AmazonComputers dataset due to computational limitations.

| Dataset | Simple | | | | | | | | Complex | | Other | | | | |
| --- | --- | --- | --- | --- | --- | --- | --- | --- | --- | --- | --- | --- | --- | --- | --- |
| | Entropy-P | | Entropy-S | | LC-P | | LC-S | | MP | ESP | ANRMAB | | AGE | | GEEM |
| | GCN | SGC | GCN | SGC | GCN | SGC | GCN | SGC | SGC | SGC | GCN | SGC | GCN | SGC | SGC |
| Citeseer | 2.03 | 3.26 | 1.82 | 4.47 | 0.68 | 2.04 | 1.55 | 3.40 | $-1.88$ | $-0.53$ | 0.17 | 0.05 | $-0.14$ | 0.16 | $-1.35$ |
| CoraML | 2.89 | 1.75 | 1.57 | 4.12 | 2.56 | 1.02 | 3.14 | 3.30 | 3.96 | 0.28 | 2.09 | 1.53 | $-0.89$ | 0.05 | $-1.77$ |
| PubMed | 4.80 | 0.52 | 4.58 | 2.58 | 5.09 | 1.06 | 6.66 | 3.17 | 3.91 | $-0.54$ | 2.71 | 0.79 | 0.81 | 0.16 | $-0.22$ |
| Amazon Photos | $-1.06$ | 1.23 | 2.20 | 5.60 | 2.47 | 0.54 | 3.98 | 3.76 | 1.95 | 0.03 | 0.87 | 0.68 | 0.12 | 0.61 | $-0.16$ |
| Amazon Computers | $-0.02$ | 0.90 | 4.52 | 5.39 | 0.84 | 3.14 | 3.21 | 5.77 | 2.61 | $-$ | $-1.53$ | $-0.70$ | $-0.09$ | 0.10 | $-$ |

runtime scaling remains consistent across CoraML (2,810 nodes, 15,962 edges) and PubMed (19,717 nodes, 88,648 edges), indicating that GATTA's computational overhead is governed by ensemble size and acquisition complexity rather than graph scale. For a detailed discussion about computational complexity, see Appendix G.

# 6 Results

This section presents the empirical evaluation of GATTA across five graph datasets, two GNN architectures, and six active learning acquisition strategies. We evaluate citation networks (Citeseer, CoraML, PubMed) and co-purchase networks (AmazonPhotos, AmazonComputers) using GCN and SGC architectures. Acquisition strategies include simple uncertainty-based methods (Least Confidence, Entropy), more sophisticated uncertainty-based methods (MP, ESP, GEEM), and structure-aware methods (AGE, ANRMAB). Unless stated otherwise, all results are averaged over 25 independent trials (5 random initial labeled pools × 5 random model initializations).

The main experiments use the same GATTA configuration identified through the sensitivity analysis in section 5: Feature Noising ($\sigma^2_{\text{noise}} = 0.4$) combined with Edge Dropout ($p_{\text{drop}} = 0.5$), an ensemble size of $N = 500$, and consistency filtering applied only to GATTA-P. Rather than selecting dataset-specific hyperparameters, we use this single configuration across all datasets, GNN architectures, and acquisition strategies. The sensitivity analysis demonstrates that GATTA consistently improves performance across a broad range of augmentation strengths and ensemble sizes, indicating that the observed gains are not tied to a single carefully tuned configuration. This motivates the use of a common default configuration rather than tuning hyperparameters separately for each dataset or acquisition strategy.

Table 2 summarizes accuracy improvements over baseline, while Figure 3 illustrates performance trajectories on CoraML. Complete statistics are reported in the Appendix (Table 7).

See more about the active learning protocol, datasets, models, and training details in Appendix A.

**R1. GATTA selectively benefits uncertainty-based methods.** Simple uncertainty methods achieve substantial improvements: Least Confidence gains +2.87% average and Entropy gains +3.03%, with GATTA-S peaks exceeding +5% on multiple datasets. Figure 3 shows GATTA-enhanced Entropy and LC exceeding baseline GEEM performance. Overall, GATTA-S outperforms GATTA-P for these methods, achieving

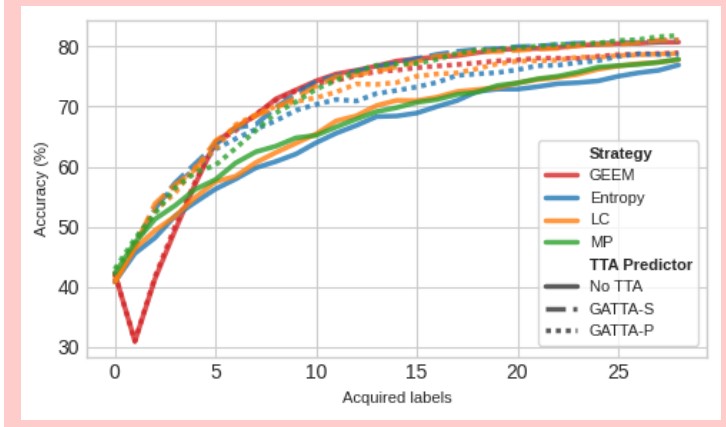

| Method | Base | GATTA-P | GATTA-S |
|---|---|---|---|
| Entropy | 76.89±4.06 | 78.64±2.83 | 81.01±1.63 |
| LC | 77.79±2.65 | 78.81±2.29 | 81.09±1.05 |
| GEEM | 80.69±1.65 | – | – |
| MP | 77.83±3.01 | 81.79±1.62 | – |

Figure 3: Learning curves across acquisition strategies on the CoraML dataset. TTA (dashed: GATTA-P, dotted: GATTA-S) improves sample efficiency for uncertainty-based strategies such as Entropy and LC, while having a limited effect on structure-based strategies like GEEM. The table reports final classification accuracy (%, mean±std) for the displayed methods; complete statistics are reported in the Appendix (Table 7).

larger and more consistent improvements by preserving per-view uncertainty through score-level aggregation. Complex methods show inconsistent results: MP achieves +2.11% average with strong gains on CoraML, PubMed, and AmazonComputers, but declines significantly on Citeseer (-1.88%). ESP averages -0.19%, declining on two datasets, suggesting potential interference between epistemic uncertainty estimation and test-time augmentation. Non-uncertainty methods derive minimal benefit: ANRMAB achieves +0.67% average while AGE shows near-zero improvement (+0.09%) and GEEM actively declines (-0.88%). Overall, methods with sophisticated uncertainty mechanisms or non-uncertainty-based selection derive inconsistent benefit from test-time augmentation.

**R2. GATTA's benefits emerge early and persist throughout learning.** Figure 3 shows that GATTA improves sample efficiency from the earliest acquisition rounds, when model uncertainty is highest and label budgets most constrained. The performance gap between GATTA-enhanced and baseline methods remains consistent across the learning trajectory, indicating that GATTA does not merely accelerate early learning but provides sustained improvement. This is particularly valuable in practical settings where labeling budgets are exhausted before model saturation. For a more detailed discussion about learning dynamics and performance, see Appendix E.

**R3. Performance depends on unique dataset characteristics, not broader graph category.** While citation and co-purchase networks achieve comparable average improvements, this aggregate masks substantial within-category variance. Among citation networks, performance ranges from strong gains on PubMed to modest improvements on Citeseer. GATTA-S with simple uncertainty methods achieves exceptional performance on co-purchase networks, with improvements exceeding +5% for both Entropy-S and LC-S on AmazonComputers using SGC. This variation suggests that specific graph properties, such as homophily, feature informativeness, or class structure, influence GATTA's effectiveness more than broad domain categories. We recommend pilot testing on a data subset before full deployment.

**R4. GATTA generalizes across architectures.** Our main experiments evaluate GATTA across six acquisition strategies using GCN and SGC. To test architectural generalizability, we additionally evaluate on GAT and GraphSAGE with Entropy acquisition. GATTA generalizes effectively across all four architectures, with both GAT and GraphSAGE showing consistent improvements on most datasets (Table 3, right). GATTA-S remains the stronger variant for GAT, while GraphSAGE exhibits more dataset-dependent behavior between GATTA-P and GATTA-S. These results confirm that GATTA functions as an architecture-agnostic module that operates at the input level, requiring no model modifications.

**R5. GATTA outperforms MC-Dropout** MC Dropout is one of the most widely used methods for uncertainty estimation in neural networks without requiring ensemble training, making it a natural base-

Table 3: (**Left**) Accuracy gains over Entropy baseline with MC Dropout (MCD) on GCN model. GATTA-P and GATTA-S consistently match or outperform MCD, while their combination yields no additive benefit. (**Right**) GATTA performance gains across GNN architectures, confirming architecture-agnostic design. Best per dataset/architecture in bold.

| GATTA | MCD | Am. Co. | Am. Ph. | Cite. | Cora | PubM. |
|---|---|---|---|---|---|---|
| − | ✓ | **+5.71** | -0.49 | +1.32 | +1.93 | +1.07 |
| P | − | -0.02 | +0.33 | +1.13 | **+2.89** | **+4.80** |
| P | ✓ | -8.39 | -0.98 | **+2.03** | +0.45 | +2.00 |
| S | − | -1.52 | **+2.20** | +1.87 | +2.09 | +3.70 |
| S | ✓ | +4.24 | -0.30 | +1.78 | +1.36 | +2.85 |

| Model | GATTA | Am. Co. | Am. Ph. | Cite. | Cora | PubM. |
|---|---|---|---|---|---|---|
| GCN | P | -0.02 | -1.06 | +2.03 | +2.89 | +4.80 |
| | S | +4.52 | +2.20 | +1.82 | +1.58 | +4.58 |
| SGC | P | +0.90 | +1.23 | +3.26 | +1.75 | +0.52 |
| | S | +5.39 | +5.60 | +4.47 | +4.12 | +2.58 |
| GAT | P | +3.24 | +3.05 | +2.83 | +0.74 | +3.81 |
| | S | +5.39 | +3.32 | +3.05 | +2.56 | +1.71 |
| SAG | P | +2.18 | -0.19 | -0.35 | -2.61 | -0.72 |
| | S | +1.93 | +1.56 | +1.66 | +0.41 | -1.21 |

line for comparison. GATTA matches or exceeds MC Dropout performance on 4 of 5 datasets, with gains up to +3.73% on PubMed (Table 3, left). However, combining GATTA with MCD yields no consistent improvement and can substantially degrade performance (GATTA-P + MCD drops performance on AmazonComputers by 8.39% below baseline). This suggests that both methods capture overlapping uncertainty information, and their combination introduces redundant or conflicting signals. Since GATTA operates at the input level without requiring architectural modifications, it offers a simpler and more effective alternative to dropout-based uncertainty estimation for graph active learning.

# 7    Discussion

Our results suggest that GATTA primarily improves the predictive distribution provided to the acquisition function rather than the acquisition objective itself. Consequently, its effectiveness depends on how strongly an acquisition strategy relies on predictive uncertainty. Acquisition functions such as Entropy and Least Confidence depend almost entirely on predictive uncertainty and therefore benefit the most from improved uncertainty estimates. In contrast, methods such as AGE, GEEM, and ESP combine uncertainty with structural information or approximations of expected model improvement. Because GATTA modifies only the uncertainty component, its relative impact is naturally smaller and may even become negative when the modified uncertainty estimates interact unfavorably with existing acquisition mechanisms. This provides a plausible explanation for the smaller gains observed for ESP and GEEM, as well as the degradation of MC Dropout on Amazon Computers.

The differing behavior of GATTA-P and GATTA-S further supports this interpretation. Graph augmentations are not guaranteed to preserve node semantics, meaning that some perturbations may introduce semantic drift rather than meaningful local uncertainty. Prediction aggregation (GATTA-P) is particularly sensitive to such perturbations because inconsistent predictions directly influence the aggregated predictive distribution before uncertainty is computed. Consistency filtering mitigates this effect by restricting aggregation to model-consistent perturbations. In contrast, score aggregation (GATTA-S) estimates uncertainty independently for each augmented view before averaging acquisition scores, making it more robust to inconsistent perturbations. This interpretation is consistent with the empirical observation that GATTA-S performs well even without filtering, whereas GATTA-P benefits substantially from consistency filtering.

These findings also have several practical implications. For practitioners, we offer four guidelines. First, prioritize simple uncertainty methods (Entropy, Least Confidence), as they benefit most from TTA and achieve performance competitive with complex baselines at substantially lower implementation effort. Second, use GATTA-S whenever computationally feasible, as it consistently outperforms GATTA-P and requires no filtering; resort to GATTA-P with filtering only when acquisition functions are expensive. Third, although GATTA performs robustly across a broad range of hyperparameter settings (section 5), we recommend moderate-to-high augmentation strengths ($\sigma^2_{\text{noise}} \in [0.4, 0.5]$, $p_{\text{drop}} \in [0.3, 0.5]$) and an ensemble size of $N = 500$, which consistently provide a favorable trade-off between performance and runtime. Finally, because the interac-

tion between test-time augmentation and acquisition strategies varies across datasets, augmentation settings should be validated on a representative subset of the target graph before large-scale deployment.

## 8 Conclusions and Limitations

This work introduced GATTA, a framework for systematically integrating test-time augmentation into graph active learning. Beyond demonstrating that test-time augmentation improves uncertainty estimation, we showed that its effectiveness depends on how uncertainty is incorporated into the acquisition process. Our empirical analysis revealed that simple uncertainty-based acquisition strategies benefit the most from improved predictive uncertainty, often matching or outperforming substantially more sophisticated methods without requiring architectural modifications or retraining. By systematically studying aggregation mechanisms, augmentation strategies, and computational trade-offs, we further derived practical deployment guidelines that characterize when and how test-time augmentation should be applied in graph active learning.

Despite these encouraging results, several limitations remain. Our evaluation focuses exclusively on transductive node classification, and extensions to link prediction, graph classification, or inductive settings remain unexplored. Furthermore, our empirical study considers five homophilic citation and co-purchase networks that reflect the standard evaluation protocol adopted in prior graph active learning work. Whether the same design principles extend to heterophilic, dynamic, or other graph domains remains an important direction for future investigation. Our consistency filtering mechanism also assumes reasonably reliable predictions on the original graph, an assumption that may not always hold during the earliest stages of active learning. Finally, while we provide a conceptual interpretation of GATTA as uncertainty estimation over a local neighborhood of graph perturbations, we leave a formal theoretical analysis of its convergence properties and sample complexity for future work.

Future work should investigate adaptive augmentation strategies that dynamically adjust perturbation strength based on model confidence or graph structure. A deeper theoretical understanding of how graph properties, such as homophily, structural sparsity, or feature informativeness, influence the interaction between test-time augmentation and uncertainty estimation would provide principled guidance for selecting augmentation strategies. The observed interactions between GATTA and more sophisticated acquisition strategies, such as ESP, GEEM, or MC Dropout, also warrant further investigation and may inspire acquisition functions designed specifically to leverage test-time augmentation.

More broadly, our findings suggest that test-time augmentation provides a general mechanism for improving uncertainty estimation in graph neural networks. Rather than introducing another task-specific acquisition strategy, GATTA establishes a graph-specific framework for integrating test-time augmentation into active learning while systematically characterizing the design choices that govern its effectiveness. By making graph active learning more accessible to practitioners while establishing TTA as a broadly applicable tool, this work opens avenues for test-time augmentation across graph machine learning

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

# A    Experimental Setup

We adopt the codebase and the experimental configuration from Fuchsgruber et al. (2024), including training procedures, datasets, and active learning protocols. Our implementation of GATTA can be found in this repository: https://anonymous.4open.science/r/gatta-8A38. Below, we describe the key components of our setup.

## A.1    Active Learning Protocol

We conduct active learning on graphs where, given an initial set of labeled nodes $\mathcal{L} \subset V$, we aim to acquire labels for unlabeled nodes $\mathcal{U} \subset V$ to maximize classifier performance. Our protocol proceeds as follows:

1. A single node is randomly drawn from each class to form the initial training set.
2. The model is initialized and trained until convergence.
3. The acquisition strategy selects an unlabeled node for labeling.
4. We add the acquired label to the training set and repeat from step (2) until the acquisition budget is exhausted.
5. After the final acquisition round, the model is retrained on all labeled nodes, and its classification accuracy is reported on the held-out test set as the final performance metric.

Following Fuchsgruber et al. (2024), we re-train the classifier from scratch after each acquisition iteration. Unless stated otherwise, we acquire one node label per iteration and fix the acquisition budget to $4C$, where $C$ is the number of classes. The resulting final training pools, therefore, contain fewer instances compared to dataset splits commonly used in standard semi-supervised learning benchmarks.

## A.2    Datasets

We evaluate our approach on standard node classification benchmark datasets from the literature. Following Fuchsgruber et al. (2024), we consider three citation networks and two co-purchase networks:

**Citation Networks:** CoraML (Getoor et al., 2005), Citeseer (Sen et al., 2008), and PubMed (Namata et al., 2012). In these datasets, nodes represent papers and edges represent citations.

**Co-purchase Networks:** AmazonComputers and AmazonPhotos (Shchur et al., 2018). In these datasets, nodes represent products and edges indicate that products are frequently co-purchased.

Dataset statistics are provided in Table 4.

Table 4: Dataset statistics. Homophily measures the fraction of edges connecting nodes of the same class.

| Dataset | #Nodes | #Edges | #Features | #Classes | Edge Density | Homophily |
|---|---|---|---|---|---|---|
| CoraML | 2,810 | 15,962 | 2,879 | 7 | 0.20% | 78.44% |
| Citeseer | 1,681 | 5,804 | 602 | 6 | 0.20% | 92.76% |
| PubMed | 19,717 | 88,648 | 500 | 3 | 0.02% | 80.24% |
| AmazonComputers | 13,381 | 491,556 | 767 | 10 | 0.27% | 77.72% |
| AmazonPhotos | 7,484 | 238,086 | 745 | 8 | 42.47% | 82.72% |

## A.3    Model Details

As hyperparameter tuning may be unrealistic in active learning settings (Regol et al., 2020), we do not perform validation-based hyperparameter optimization for our models. Instead, we adopt hyperparameters reported as effective in the literature and apply them uniformly across all datasets. Specifically, we use the configuration for both SGC (Wu et al., 2019a) and GCN (Kipf & Welling, 2016) from Table 5.

Table 5: Hyperparameters for GNN models.

| Layers | Hidden Dim. | Learning Rate | Max Epochs | Weight Decay | Dropout |
|--------|-------------|---------------|------------|--------------|---------|
| 1 | [64] | 0.001 | 10,000 | 0.001 | 0.8 |

### A.4 Training and Evaluation Details

We train all models using the binary cross-entropy loss with the Adam optimizer (Kingma & Ba, 2014), learning rate of $10^{-3}$, and weight decay of $10^{-3}$. We perform early stopping on validation loss with patience of 100 iterations.

For each dataset, acquisition function, and model, we evaluate five dataset splits with five independent model initializations each (25 runs in total), and report the averaged results. A priori, we fix 20% of all nodes as a test set that is reused across all splits and initializations and cannot be acquired by any strategy. For each dataset split, we fix 20% of all nodes as a validation set and reuse it across initializations.

We report test accuracy as our primary performance metric. All reported accuracy gains are computed with respect to the corresponding non-TTA baseline, i.e., the same dataset-model-acquisition configuration evaluated without test-time augmentation.

## B Information-Theoretic Interpretation of GATTA Aggregation

The theoretical framework presented in Section 3 applies to arbitrary uncertainty-based acquisition functions. The following decomposition provides insight into why prediction aggregation (GATTA-P) is empirically more sensitive to non-label-preserving graph augmentations than score aggregation (GATTA-S) for entropy-based acquisition.

Let $p_\varphi(y) = f_\theta(\varphi(G))$ denote the predictive distribution obtained from an augmented graph sampled according to $\varphi \sim q(\varphi \mid G)$, and let $\bar{p}(y) = \mathbb{E}_\varphi[p_\varphi(y)]$ denote the corresponding averaged predictive distribution. For entropy-based acquisition, where $Q(p) = H(p)$, GATTA-S estimates the expected entropy $\mathbb{E}_\varphi[H(p_\varphi)]$, whereas GATTA-P computes the entropy of the averaged predictive distribution $H(\bar{p})$. These quantities are related through the identity

$$H(\bar{p}) = \mathbb{E}_\varphi[H(p_\varphi)] + I(Y; \varphi \mid G), \tag{7}$$

where $I(Y; \varphi \mid G) = H(\bar{p}) - \mathbb{E}_\varphi[H(p_\varphi)]$ is the mutual information between the predicted class $Y$ and the sampled graph augmentation $\varphi$, conditioned on the observed graph $G$.

Equation equation 7 shows that the two aggregation strategies estimate different quantities. GATTA-S estimates the first term, corresponding to the average uncertainty of individual augmented graph realizations. GATTA-P additionally incorporates the disagreement term $I(Y; \varphi \mid G)$, which measures the variability of predictions across augmented graph views.

In Bayesian active learning disagreement between stochastic model samples is commonly interpreted as epistemic uncertainty (Gal et al., 2017). In GATTA, however, disagreement arises from stochastic graph perturbations rather than uncertainty over model parameters. Since graph augmentations are not guaranteed to preserve node semantics, disagreement between augmented graph views may reflect both meaningful local prediction instability and semantic drift introduced by the augmentation process.

By averaging predictive distributions, GATTA-P explicitly incorporates the disagreement term $I(Y; \varphi \mid G)$. When disagreement is dominated by non-label-preserving perturbations rather than informative local uncertainty, GATTA-P may overestimate predictive uncertainty, leading to less reliable acquisition scores. Consistency filtering mitigates this effect by approximately restricting aggregation to model-consistent perturbations. In contrast, GATTA-S estimates only the expected entropy of individual augmented graph re-

alizations and is therefore inherently less sensitive to disagreement between otherwise confident predictions across augmented views.

## C    Filtering variants

The filtering mechanism described in Section 3.5, enforces strict prediction consistency through binary masks, which we refer to as *Hard filtering* from now on. We additionally explored a confidence-weighted variant motivated by the intuition that not all consistent predictions should contribute equally. Instead of binary masks, we compute soft weights based on each view's confidence for the original prediction's class:

$$s_i^{(v)} = (\mathbf{P}_i)_{v,c^\star}, \quad \text{where} \quad c^\star = \arg\max_c(\mathbf{P}_0)_{v,c} \tag{8}$$

Using these soft weights alone proved ineffective, as inconsistent views with spuriously high confidence for the wrong class introduced noise. However, combining hard and soft filtering, discarding inconsistent views while weighting consistent ones by confidence, yielded a viable alternative we term *Firm filtering*:

$$w_i^{(v)} = m_i^{(v)} \cdot s_i^{(v)} \tag{9}$$

Since $m_i^{(v)}$ masks out inconsistent predictions, firm filtering reduces to weighting each view by its confidence for the predicted class. Despite early promising results, firm filtering did not outperform hard filtering in our experiments (Table 6), suggesting that uniform weighting of consistent views is sufficient. We report firm filtering results in Figures 11–14 for completeness.

Table 6: Performance comparison (% accuracy gain over baseline) for GATTA variants under different filtering strategies. Results averaged across CoraML and PubMed with GCN and SGC models using Entropy and LC.

| Filtering | GATTA-P | GATTA-S | *Average* |
|-----------|---------|---------|-----------|
| Hard | $1.96 \pm 1.27$ | $2.16 \pm 1.28$ | $2.06 \pm 1.28$ |
| Firm | $1.97 \pm 1.34$ | $2.14 \pm 1.35$ | $2.06 \pm 1.34$ |
| None | $0.12 \pm 1.04$ | $2.49 \pm 1.42$ | $1.31 \pm 1.72$ |
| *Average* | $1.37 \pm 1.5$ | $2.26 \pm 1.36$ | $1.81 \pm 1.50$ |

## D    Impact of Data Augmentation

In this section, we provide a detailed analysis of augmentation strength experiments for Feature Noising and Edge Drop across two datasets (CoraML, PubMed) and two acquisition strategies (Entropy, LC), as referenced in Section 5.2. Figures 11– 14 report accuracy gains relative to the corresponding non-TTA baseline, that is, the same dataset-model-acquisition configuration evaluated without test-time augmentation.

As discussed in Section 5.2, stronger augmentations generally yield better performance. Notably, filtering mechanisms play distinct roles depending on the aggregation strategy: for GATTA-P, consistency-based filtering is essential to maintain performance, particularly at higher augmentation strengths, whereas GATTA-S achieves optimal results without filtering. These trends are consistent across both SGC and GCN architectures. Additionally, we observe no substantial performance difference between Firm and Hard filtering variants.

## E    Learning Dynamics and Performance Comparison

Figures 5– 9 present learning curves across acquisition strategies, GNN architectures, and datasets. The upper row displays results for GCN, while the bottom row shows SGC performance. The left column

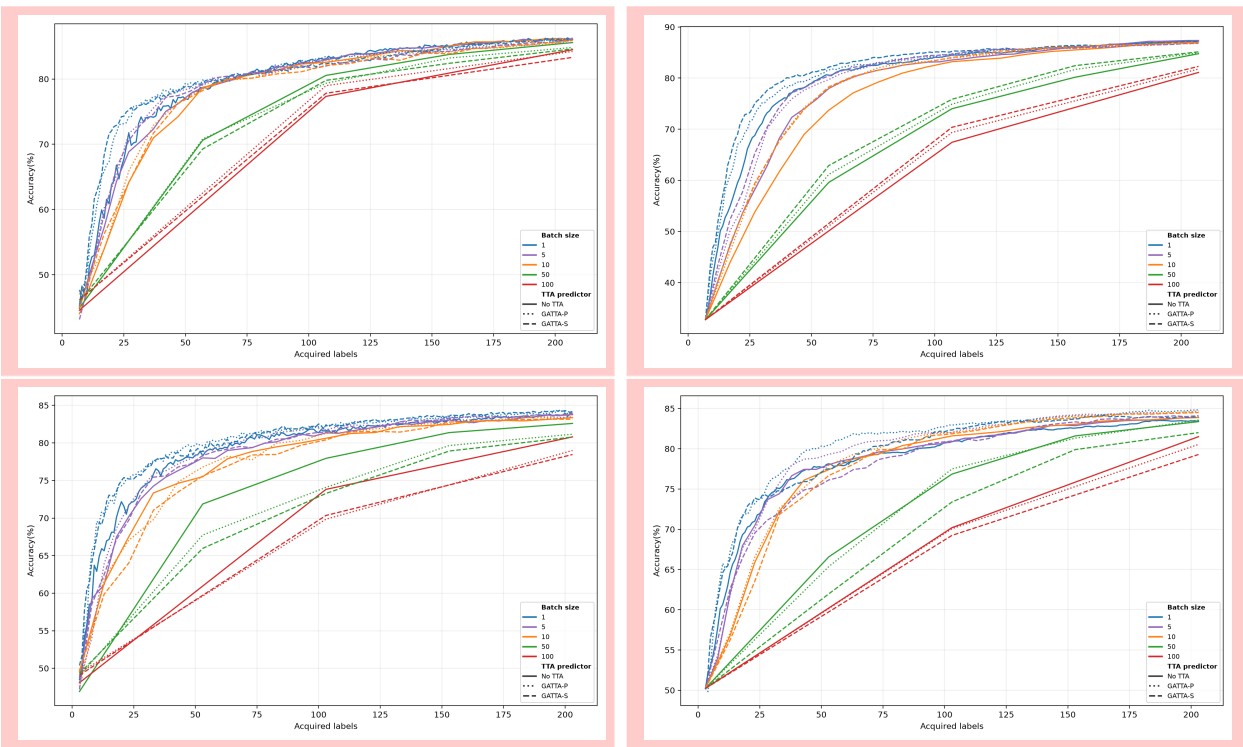

Figure 4: **Effect of acquisition batch size and labeling budget.** Learning curves for GATTA-S, GATTA-P, and the corresponding baseline acquisition strategy under acquisition batch sizes of 1, 5, 10, 50, and 100 nodes per iteration. Results are shown for the CoraML (top row) and PubMed (bottom row) datasets using GCN (left column) and SGC (right column). Curves are aligned by the total number of acquired labels. Larger acquisition batches reduce the benefit of improved uncertainty estimation, with GATTA providing the largest gains during the early stages of active learning.

presents simple uncertainty-based strategies (Entropy, LC), and the right column shows complex strategies (AGE, ANRMAB, ESP, MP, GEEM). Following the original implementations (Fuchsgruber et al., 2024; Regol et al., 2020), ESP, MP, and GEEM were evaluated exclusively with SGC. Table 7 summarizes the final test accuracies after all active learning iterations, providing a quantitative complement to the learning curve visualizations.

Both GATTA variants substantially improve simple strategies, enabling them to close the performance gap with GEEM and, in several cases, surpass it (Citeseer, CoraML, PubMed). The improvements are most pronounced for simple uncertainty-based methods and, notably, for the MP strategy. Across architectures, GATTA-P and GATTA-S demonstrate consistent benefits, with GATTA-S often achieving better performance in later iterations, particularly on citation networks.

Dataset characteristics significantly influence GATTA's effectiveness, though not along simple domain boundaries. Within citation networks, performance ranges from strong gains on PubMed to modest improvements on Citeseer. Co-purchase networks show high variance: Amazon Computers achieves exceptional improvements (+5% or more with GATTA-S), while Amazon Photos shows more moderate gains. This variation suggests that specific graph properties—such as feature informativeness, homophily, or class balance—matter more than broad dataset categories.

For complex strategies like AGE and ANRMAB, GATTA provides more modest improvements and, in some cases, may degrade performance. This suggests that sophisticated acquisition functions already incorporate mechanisms that partially account for prediction uncertainty, making additional augmentation-based refinement less beneficial. Similarly, GEEM shows consistent slight degradation with GATTA, indicating potential interference between structure-aware acquisition and input-level perturbations.

Beyond accuracy improvements, GATTA consistently reduces performance variance. Across all simple method configurations (Entropy and LC with GCN and SGC), GATTA-S reduces standard deviation in 18 of 20 dataset-architecture combinations, with the exceptions occurring on PubMed with SGC. This indicates more reliable uncertainty estimation: aggregating predictions across augmented views stabilizes node selection, yielding more consistent outcomes across initializations.

# F    Batch Active Learning and Extended Labeling Budgets

We further investigated the effect of acquisition batch size by increasing the number of queried nodes per active learning iteration. Figure 4 illustrates the effect of the acquisition batch size (1, 5, 10, 50, and 100 queried nodes per iteration) across the CoraML and PubMed datasets using both GCN and SGC architectures. To enable direct comparison, the learning curves are aligned by the total number of acquired labels. As expected, larger acquisition batches consistently degraded performance for both the baseline and GATTA-enhanced acquisition strategies, reflecting the reduced opportunity to update the model between successive label acquisitions.

The benefit of GATTA also decreased with increasing batch size. For sequential or small-batch acquisition, refining the predictive uncertainty leads to improved node rankings and consistently higher downstream accuracy. As the batch size increases, however, acquisition decisions depend on a much larger portion of the uncertainty ranking, reducing the relative advantage of improved uncertainty estimation. A possible explanation is that larger batches contain increasingly similar uncertain nodes, leading to more redundant acquisitions despite improved uncertainty estimates.

Finally, as shown in Figure 4, the performance gap between GATTA and the baseline is largest during the early stages of active learning and gradually diminishes as more labels are acquired. This observation is consistent with the intuition that uncertainty estimation is most valuable when labeled data are scarce, whereas acquisition strategies naturally converge as the labeling budget increases.

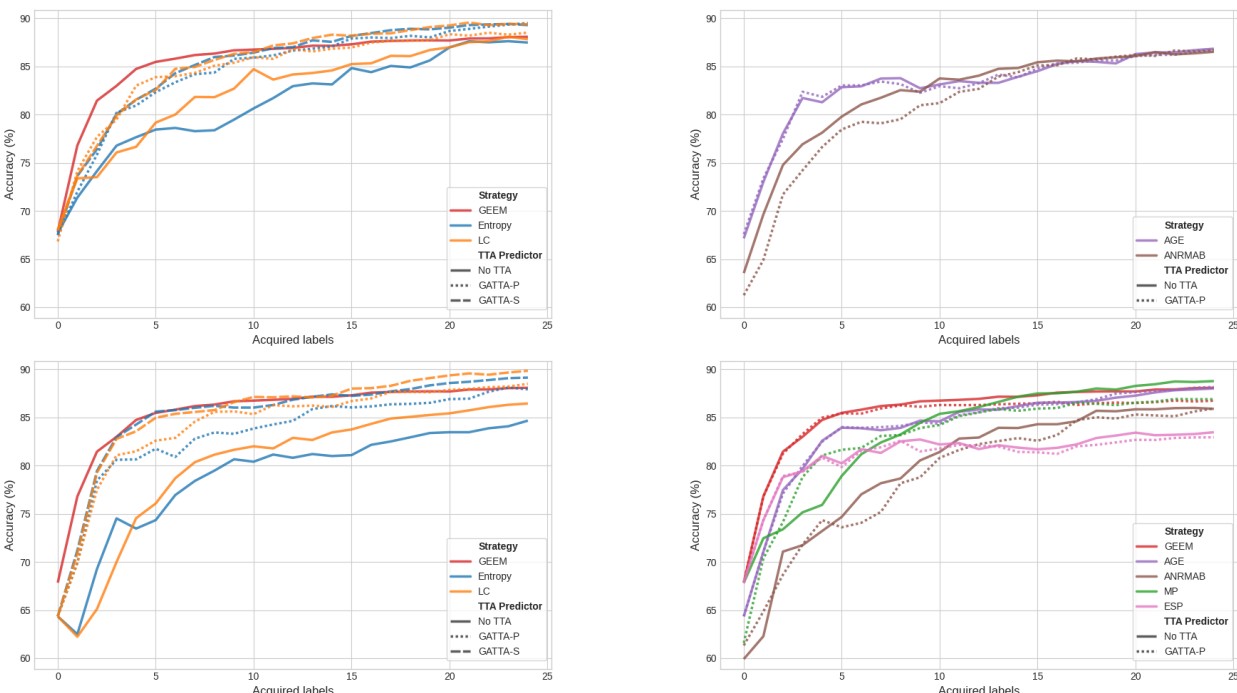

Figure 5: Active learning curves for Citeseer across acquisition strategies and architectures. The rows show performance for GCN (top) and SGC (bottom).

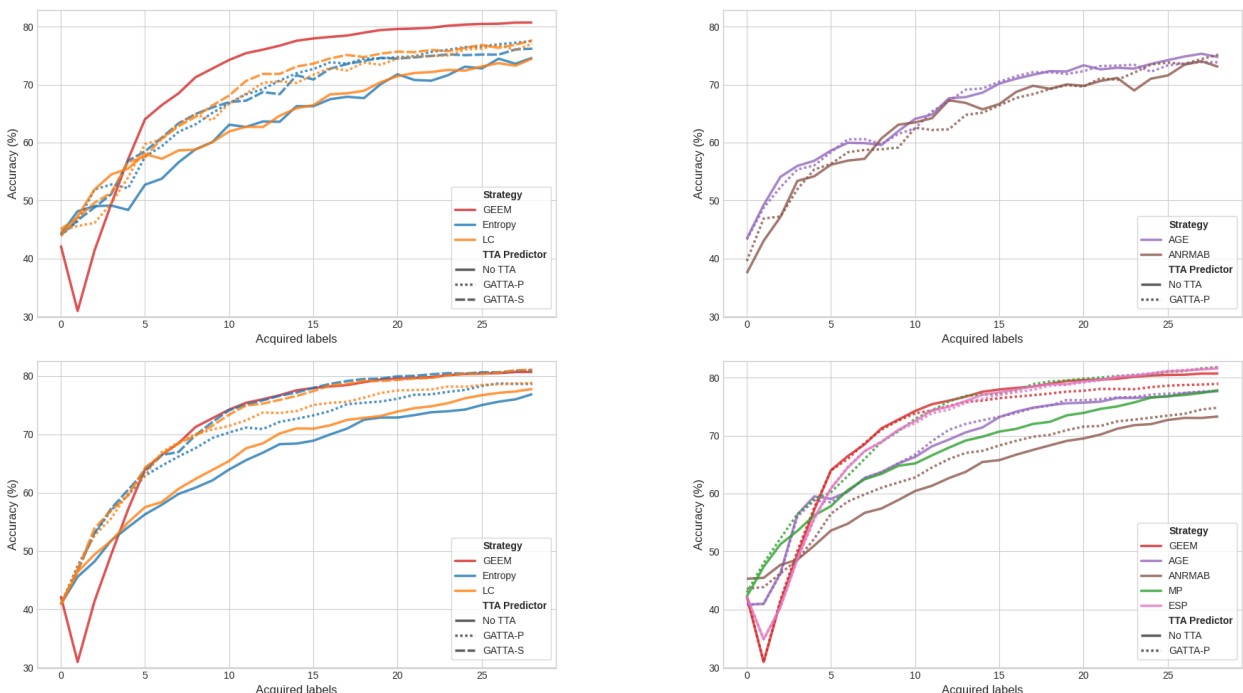

Figure 6: Active learning curves for CoraML across acquisition strategies and architectures. The rows show performance for GCN (top) and SGC (bottom).

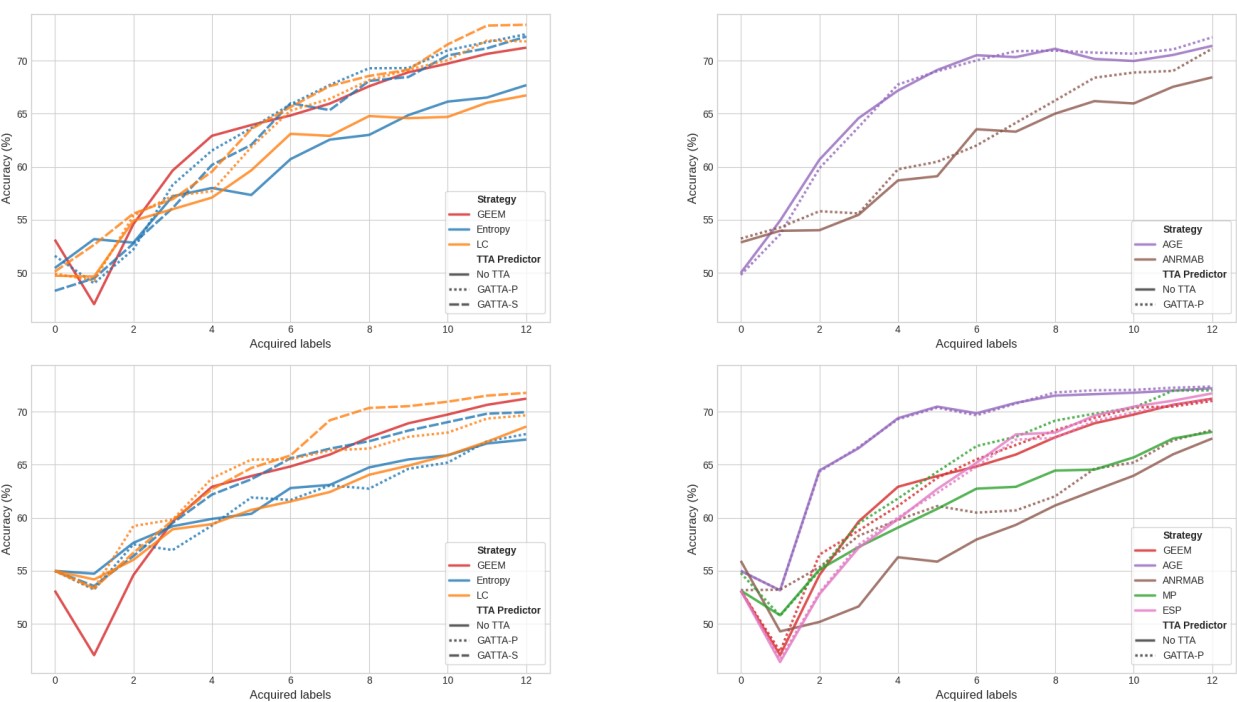

Figure 7: Active learning curves for PubMed across acquisition strategies and architectures. The rows show performance for GCN (top) and SGC (bottom).

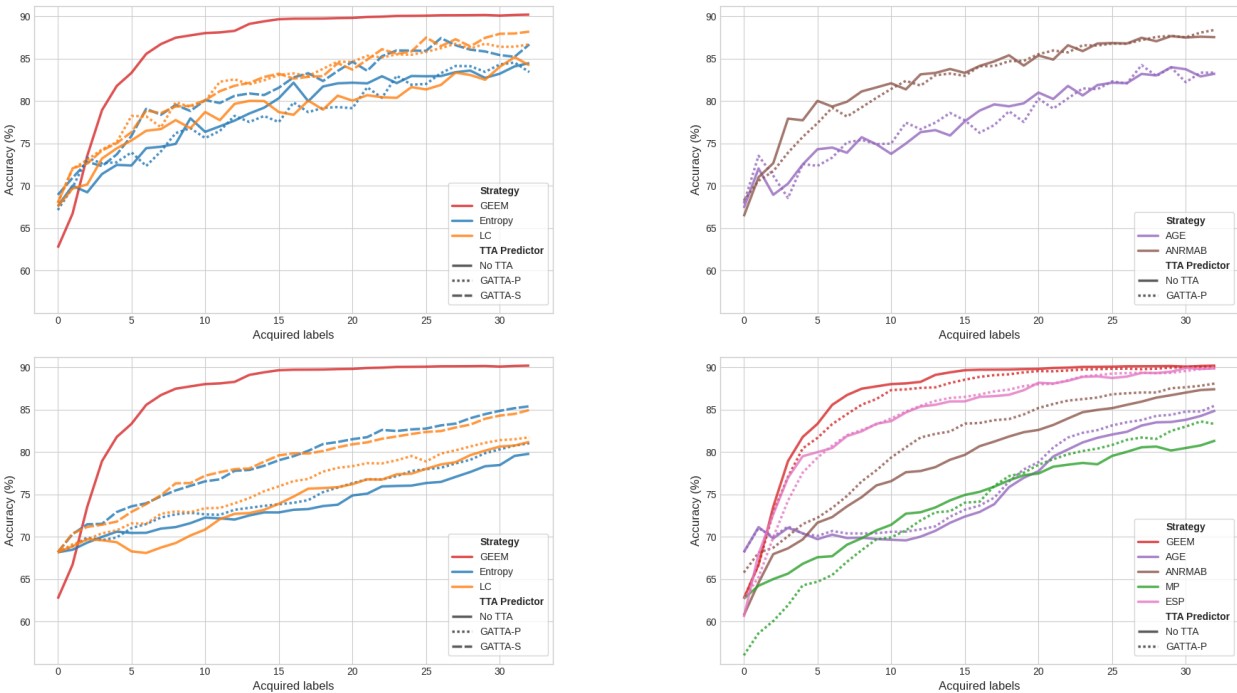

Figure 8: Active learning curves for Amazon Photos across acquisition strategies and architectures. The rows show performance for GCN (top) and SGC (bottom).

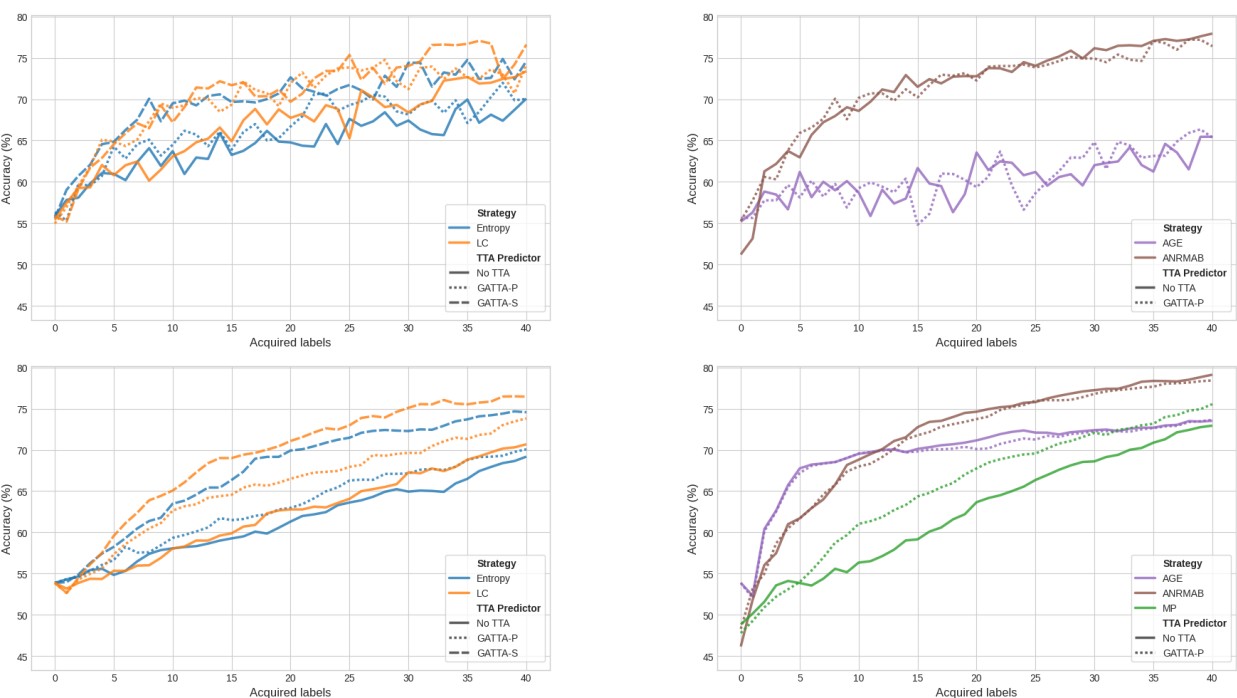

Figure 9: Active learning curves for Amazon Computers across acquisition strategies and architectures. The rows show performance for GCN (top) and SGC (bottom).

Table 7: Test accuracy (%) across datasets, acquisition strategies, and GNN architectures. Results compare baseline strategies (no GATTA indicator) with GATTA-P and GATTA-S variants using GCN and SGC architectures. Bold indicates best performance per strategy-dataset combination. Standard deviations computed over 25 (5 dataset initializations × 5 model initializations) runs. Missing entries indicate experiments not conducted for that configuration due to computational limitations.

| Strategy | GATTA | Citeseer | | CoraML | | PubMed | | Amazon Photos | | Amazon Computers | |
|---|---|---|---|---|---|---|---|---|---|---|---|
| | | GCN | SGC | GCN | SGC | GCN | SGC | GCN | SGC | GCN | SGC |
| Entropy | – | 87.48±3.37 | 84.67±6.29 | 74.61±4.61 | 76.89±4.06 | 67.69±6.01 | 67.37±5.79 | 84.47±5.99 | 79.79±8.91 | 70.06±7.81 | 69.18±6.63 |
| | P | 89.51±1.60 | 87.93±1.81 | 77.50±2.67 | 78.64±2.83 | 72.49±3.53 | 67.89±7.98 | 83.41±6.34 | 81.02±7.98 | 70.04±6.91 | 70.08±7.24 |
| | S | 89.30±1.35 | 89.14±1.22 | 76.19±3.35 | 81.01±1.63 | 72.27±5.49 | 69.95±6.75 | 86.67±3.83 | 85.39±5.99 | 74.58±6.51 | 74.57±4.20 |
| LC | – | 87.83±2.26 | 86.45±2.61 | 74.49±4.05 | 77.79±2.65 | 66.72±7.27 | 68.59±4.92 | 84.20±5.93 | 81.19±7.66 | 73.38±5.47 | 70.68±6.64 |
| | P | 88.52±2.11 | 88.49±1.50 | 77.05±3.63 | 78.81±2.29 | 71.81±3.85 | 69.65±8.77 | 86.67±4.67 | 81.73±9.05 | 74.22±3.88 | 73.82±6.26 |
| | S | 89.38±1.37 | 89.85±1.16 | 77.63±2.34 | 81.09±1.05 | 73.38±3.26 | 71.77±5.87 | 88.18±3.85 | 84.95±5.37 | 76.59±4.35 | 76.45±4.77 |
| ANRMAB | – | 86.52±1.91 | 85.90±2.91 | 73.06±5.15 | 73.29±5.02 | 68.43±7.41 | 67.47±6.03 | 87.53±2.48 | 87.41±2.46 | 77.93±3.01 | 79.12±2.25 |
| | P | 86.69±1.83 | 85.95±3.34 | 75.15±4.9 | 74.81±4.42 | 71.13±5.63 | 68.26±5.45 | 88.40±2.55 | 88.08±2.58 | 76.40±3.22 | 78.42±2.59 |
| AGE | – | 86.85±1.94 | 88.01±1.55 | 74.73±2.79 | 77.64±1.83 | 71.4±4.85 | 72.20±6.87 | 83.23±4.29 | 84.88±3.48 | 65.46±6.06 | 73.53±4.11 |
| | P | 86.70±1.56 | 88.17±1.27 | 73.84±1.74 | 77.69±1.59 | 72.20±4.79 | 72.37±6.03 | 83.34±3.92 | 85.49±3.84 | 65.37±7.56 | 73.63±3.76 |
| GEEM | – | – | 88.08±1.00 | – | 80.69±1.65 | – | 71.22±4.87 | – | 90.19±1.38 | – | – |
| | P | – | 86.74±1.89 | – | 78.92±2.97 | – | 71.01±3.59 | – | 90.04±1.27 | – | – |
| MP | – | – | 88.78±1.56 | – | 77.83±3.01 | – | 68.09±6.94 | – | 81.35±9.33 | – | 72.94±7.66 |
| | P | – | 86.89±2.92 | – | 81.79±1.62 | – | 72.01±4.41 | – | 83.29±5.35 | – | 75.55±7.45 |
| ESP | – | – | 83.47±2.36 | – | 81.55±1.77 | – | 71.72±5.43 | – | 89.87±1.29 | – | – |
| | P | – | 82.94±2.77 | – | 81.83±1.98 | – | 71.18±5.67 | – | 89.89±1.80 | – | – |

## G   Computational complexity

Figure 2 presents the computational overhead for a single active learning iteration across acquisition strategies, datasets, and GATTA variants. The runtime patterns remain consistent across datasets, revealing critical differences in computational scaling between aggregation approaches.

For simple strategies like Entropy, both GATTA-P and GATTA-S exhibit similar runtime scaling, with overhead growing modestly with the number of augmentations. This pattern extends to complex strategies when using GATTA-P, which maintains comparable runtime regardless of acquisition function complexity. In contrast, GATTA-S with complex strategies (e.g., AGE) exhibits substantially higher computational costs that scale linearly with both the number of augmentations and the acquisition function's complexity.

This disparity stems from the algorithmic differences. GATTA-P has computational complexity $\mathcal{O}((N \times I) + Q)$, where $N$ is the number of augmentations, $\mathcal{O}(I)$ is the inference cost per augmentation, and $\mathcal{O}(Q)$ is the acquisition function evaluation cost. The acquisition function is computed only once on aggregated predictions. Conversely, GATTA-S has complexity $\mathcal{O}(N \times (I + Q))$, requiring $N + 1$ separate evaluations of the acquisition function—once per augmented view.

When $\mathcal{O}(Q)$ is negligible compared to $\mathcal{O}(I)$ (as with simple strategies like Entropy or LC), both variants exhibit similar runtimes. However, when $\mathcal{O}(Q)$ dominates, as with complex strategies like AGE that require expensive graph computations, GATTA-S incurs a multiplicative overhead of $N \times \mathcal{O}(Q)$, resulting in dramatically increased iteration times. For instance, at 10,000 augmentations with AGE on PubMed, GATTA-S requires over 5,000 seconds per iteration compared to approximately 50 seconds for GATTA-P, representing a 100-fold difference.

These results suggest a clear practical guideline: GATTA-P is preferable for complex acquisition functions due to its computational efficiency, while GATTA-S may be suitable for simple strategies where the additional repeated evaluations impose minimal overhead and potentially provide marginal performance benefits.

## H   Confidence Analysis

Figure 10 illustrates the evolution of prediction confidence distributions across 25 active learning iterations for the Entropy strategy on Cora-ML with GATTA-P, averaged over 25 runs. Each panel displays a two-dimensional histogram where the y-axis represents confidence levels (0.0 to 1.0), the x-axis shows iteration number, color intensity indicates the frequency of nodes at each confidence level, and the red line traces mean confidence over iterations.

The baseline confidences exhibit a wide distribution throughout the active learning process. This broad dispersion reflects the inherent uncertainty in the model's predictions on unlabeled nodes. In contrast, GATTA confidences averaged over 500 augmented views (top right panel) show a more concentrated distribution, with increased density in the mid-range confidence region (0.4-0.6). This concentration effect suggests that test-time augmentation reveals underlying prediction uncertainty by reducing both overconfident and underconfident predictions, moderating them toward more calibrated estimates.

Notably, the mean confidence remains consistently above the most concentrated region in both the original and GATTA distributions, indicating that the distribution is skewed toward higher confidences. This asymmetry is characteristic of uncertainty-based active learning, where the strategy progressively queries uncertain nodes, leaving a larger proportion of high-confidence predictions in the unlabeled pool.

The filtered GATTA confidences demonstrate the effect of consistency-based filtering. Compared to unfiltered GATTA, filtering shifts the distribution upward, reducing the density of low-confidence predictions ($<0.4$). This elevation effect occurs because filtering discards augmented views with inconsistent predictions, retaining only the more stable, higher-confidence predictions. The result is a distribution that is both more concentrated and shifted toward higher confidence values, suggesting improved calibration through the removal of unreliable augmentation-induced predictions.

These distributional changes have direct implications for active learning: the concentration and elevation effects indicate that GATTA provides more reliable uncertainty estimates, potentially leading to better node selection and improved label efficiency.

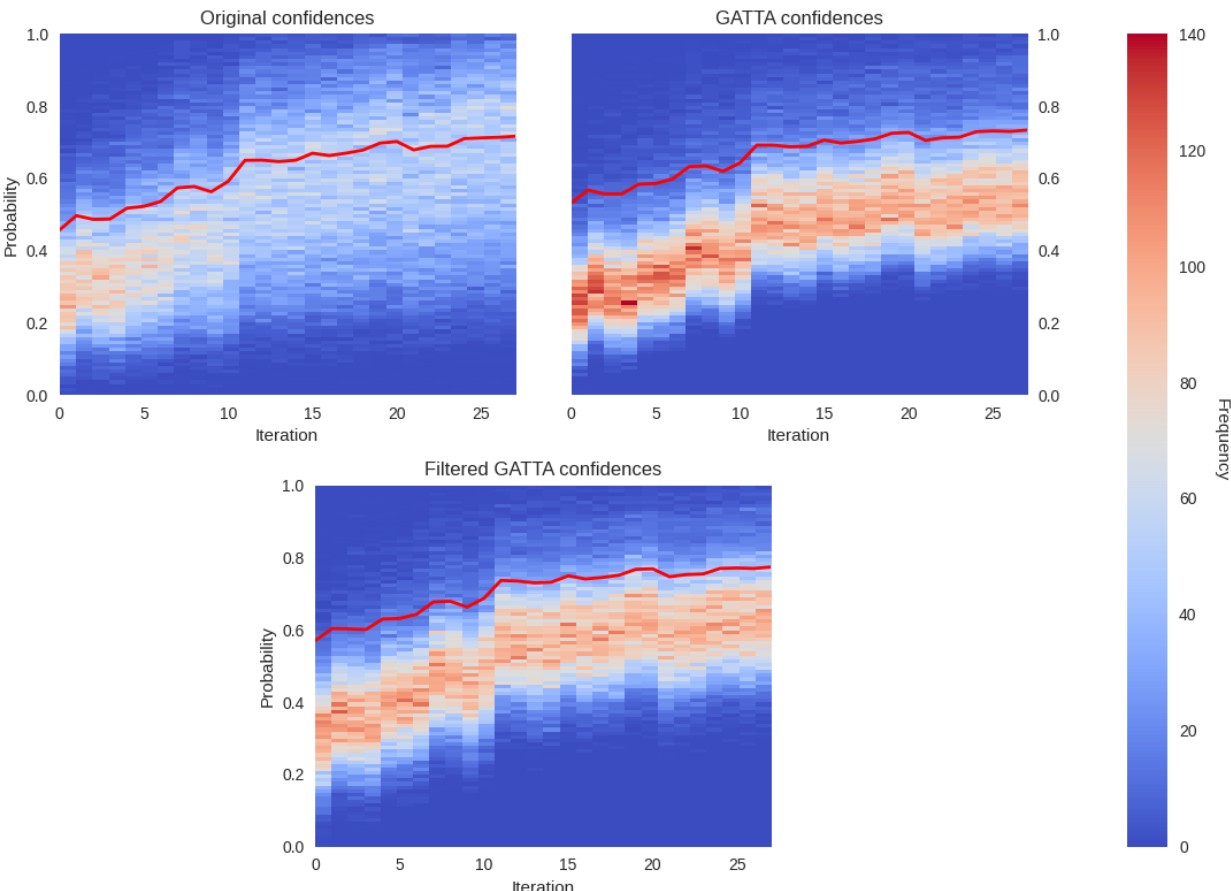

Figure 10: Confidence distribution evolution across active learning iterations for Entropy on Cora-ML with GATTA-P. Heat maps show the distribution of prediction confidences (y-axis) over active learning iterations (x-axis), averaged over 25 runs. Top left: baseline confidences from the original graph. Top right: confidences averaged over 500 augmented views. Bottom: confidences after applying consistency-based filtering. The red line indicates mean confidence per iteration. Color intensity represents the frequency of nodes at each confidence level.

Table 8: Comparison of GATTA variants with MC Dropout (MCD) on Entropy-based acquisition. GATTA-P and GATTA-S consistently match or outperform MCD, while their combination yields no additive benefit. Best results per dataset in bold.

|  | Amazon Computers | Amazon Photos | CiteSeer | CoraML | PubMed |
|---|---|---|---|---|---|
| Entropy | 70.06 ± 7.97 | 84.47 ± 6.12 | 87.48 ± 3.44 | 74.61 ± 4.7 | 67.69 ± 6.13 |
| + MCD | **75.77 ± 3.64** | 83.98 ± 4.41 | 88.8 ± 1.72 | 76.54 ± 3.62 | 68.76 ± 5.63 |
| GATTA-P | 70.04 ± 7.05 | 84.8 ± 6.13 | 88.61 ± 1.75 | **77.5 ± 2.72** | **72.49 ± 3.6** |
| + MCD | 61.67 ± 9.31 | 83.49 ± 5.83 | **89.51 ± 1.07** | 75.06 ± 3.11 | 69.69 ± 5.61 |
| GATTA-S | 68.54 ± 10.98 | **86.67 ± 3.91** | 89.35 ± 1.01 | 76.7 ± 2.83 | 71.39 ± 4.58 |
| + MCD | 74.3 ± 5.5 | 84.17 ± 7.42 | 89.26 ± 1.45 | 75.97 ± 2.74 | 70.54 ± 4.74 |

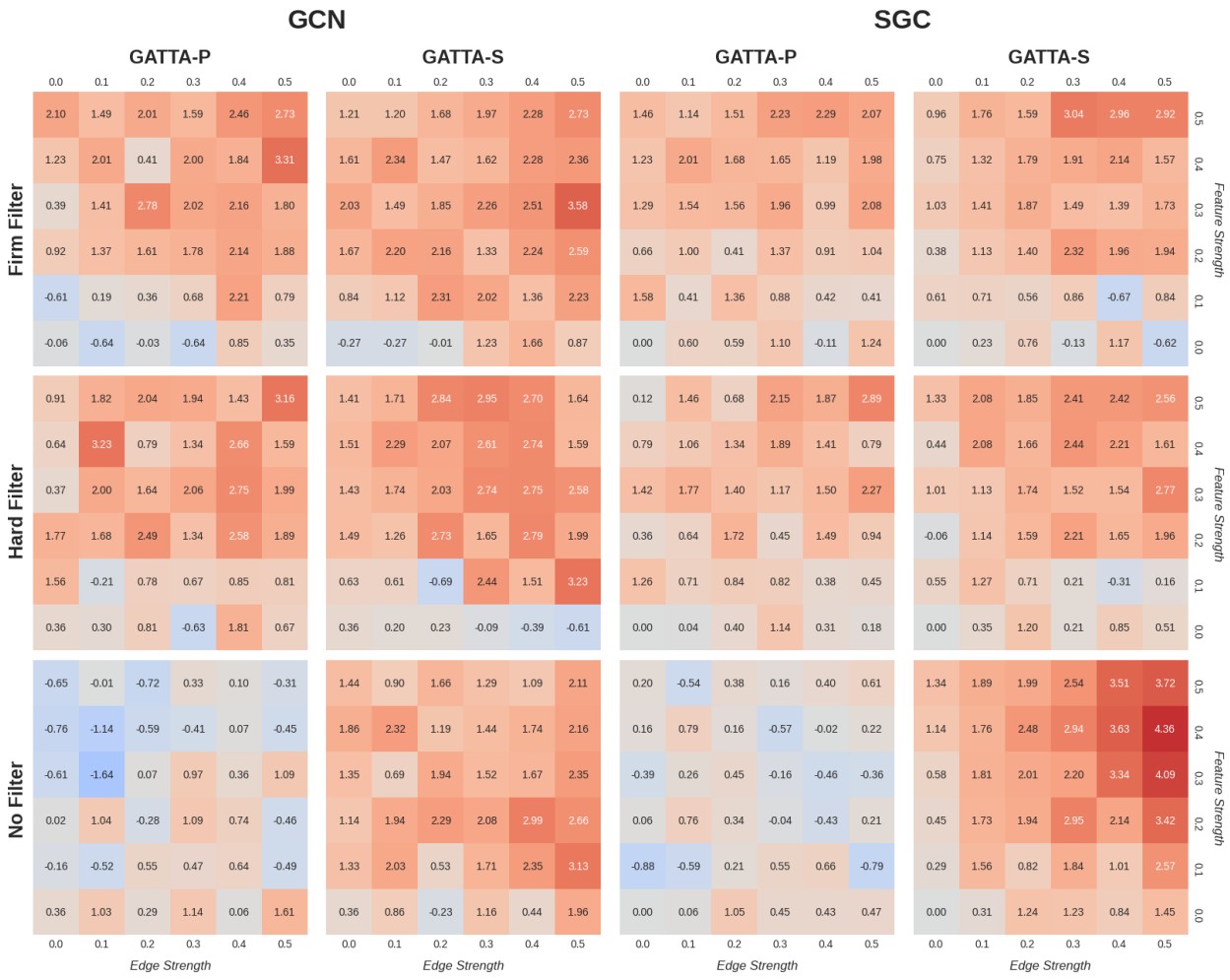

Figure 11: Performance sensitivity to augmentation strength and filtering for **Entropy** on **CoraML**. Heatmaps show performance improvement (%) relative to baseline for GATTA-P and GATTA-S with GCN and SGC architectures. Rows correspond to Firm Filter, Hard Filter, and No Filter. Axes represent edge dropout (horizontal) and feature noising (vertical) strengths from 0.0 to 0.5. Darker red indicates larger improvements; blue indicates degradation.

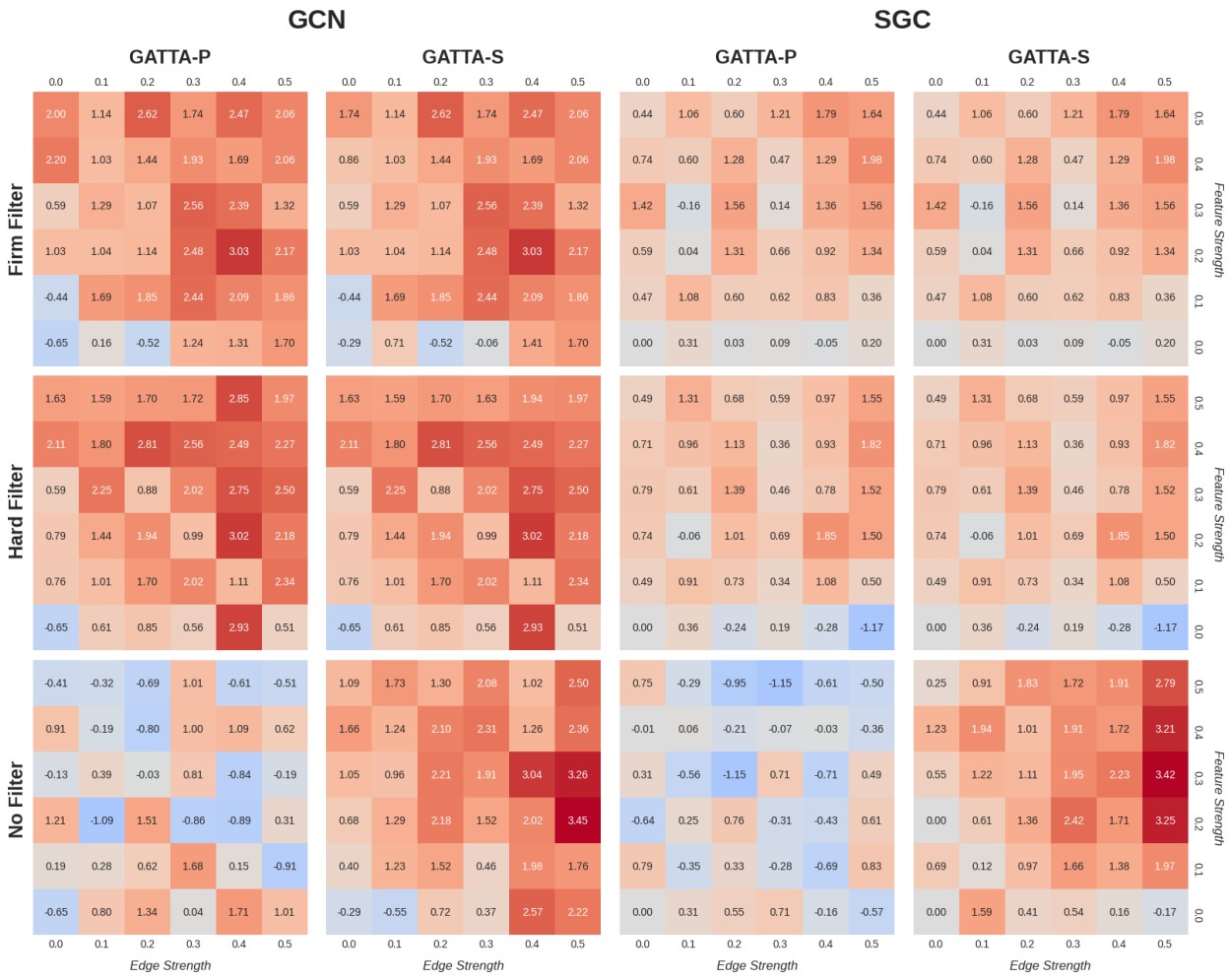

Figure 12: Performance sensitivity to augmentation strength and filtering for **LC** on **CoraML**. Heatmaps show performance improvement (%) relative to baseline for GATTA-P and GATTA-S with GCN and SGC architectures. Rows correspond to Firm Filter, Hard Filter, and No Filter. Axes represent edge dropout (horizontal) and feature noising (vertical) strengths from 0.0 to 0.5. Darker red indicates larger improvements; blue indicates degradation.

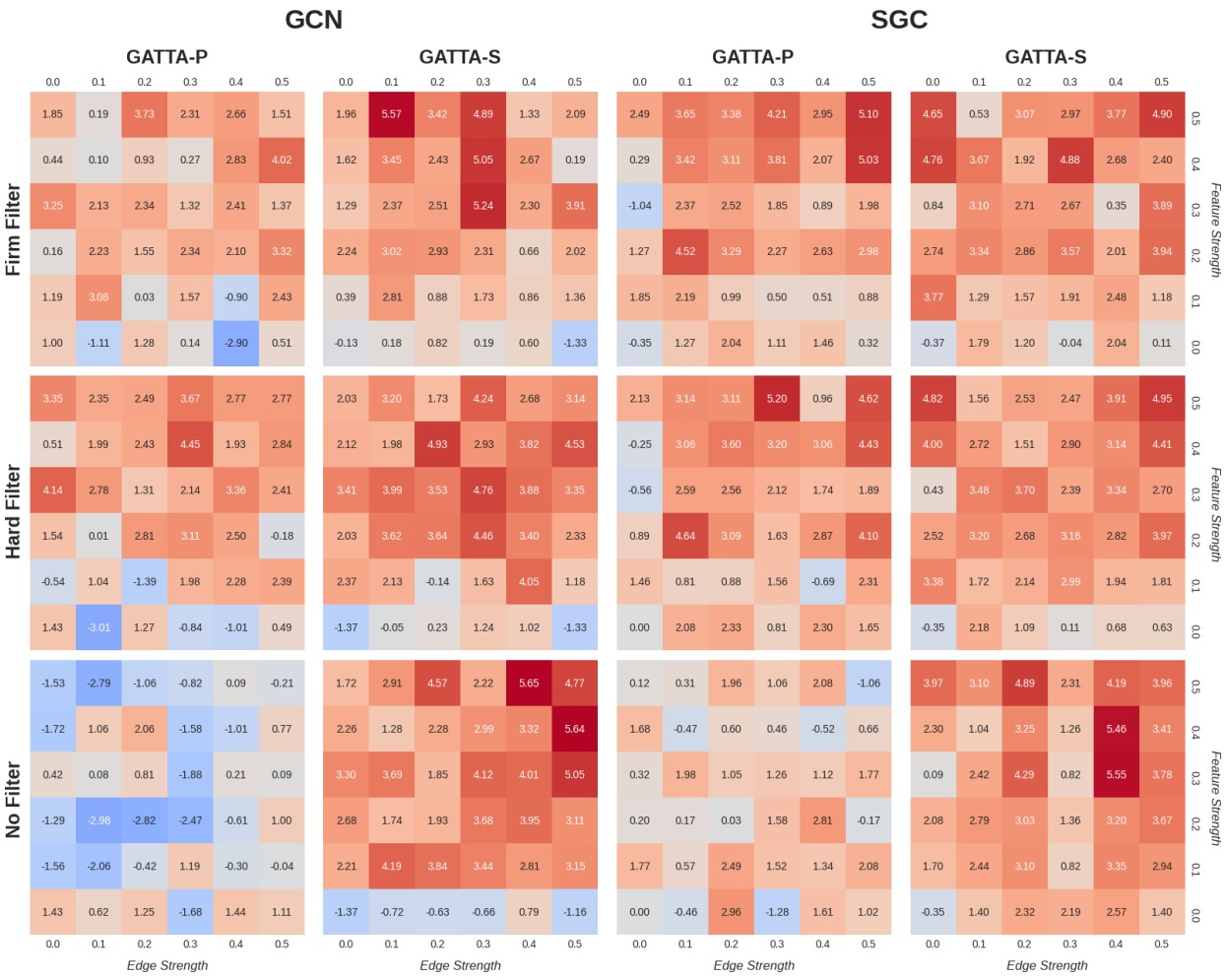

Figure 13: Performance sensitivity to augmentation strength and filtering for **Entropy** on **PubMed**. Heatmaps show performance improvement (%) relative to baseline for GATTA-P and GATTA-S with GCN and SGC architectures. Rows correspond to Firm Filter, Hard Filter, and No Filter. Axes represent edge dropout (horizontal) and feature noising (vertical) strengths from 0.0 to 0.5. Darker red indicates larger improvements; blue indicates degradation.

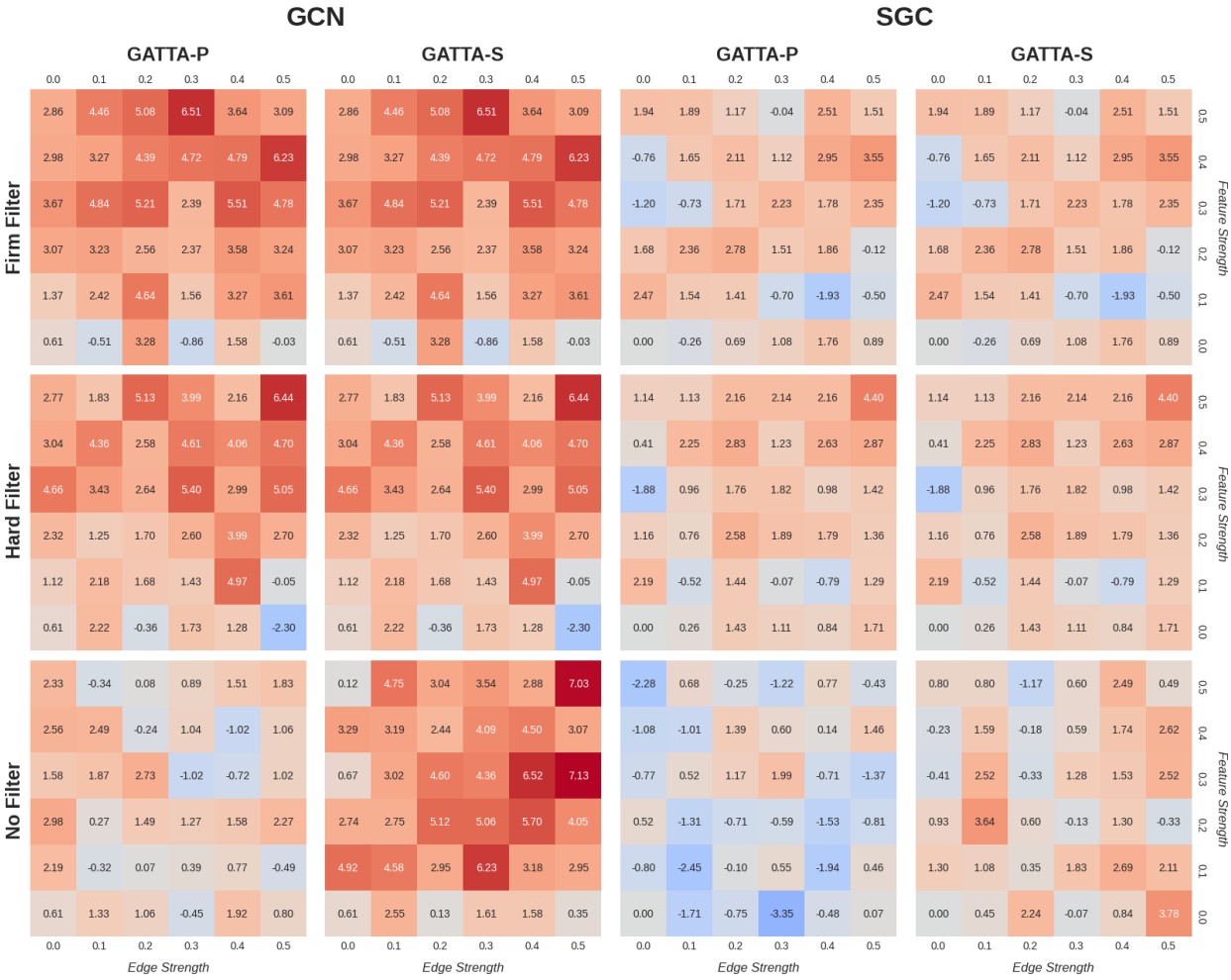

Figure 14: Performance sensitivity to augmentation strength and filtering for **LC** on **PubMed**. Heatmaps show performance improvement (%) relative to baseline for GATTA-P and GATTA-S with GCN and SGC architectures. Rows correspond to Firm Filter, Hard Filter, and No Filter. Axes represent edge dropout (horizontal) and feature noising (vertical) strengths from 0.0 to 0.5. Darker red indicates larger improvements; blue indicates degradation.

Table 9: GATTA performance across GNN architectures with Entropy strategy. Rows marked "–" denote baseline (no GATTA), "P" denotes GATTA-P with filtering, and "S" denotes GATTA-S without filtering. All four architectures show consistent improvements over baselines, confirming GATTA's architecture-agnostic design. Best results per architecture and dataset in bold.

| | | Amazon Computers | Amazon Photos | CiteSeer | CoraML | PubMed |
|---|---|---|---|---|---|---|
| GCN | – | $70.06 \pm 7.81$ | $84.47 \pm 5.99$ | $87.48 \pm 3.37$ | $74.61 \pm 4.61$ | $67.69 \pm 6.01$ |
| | P | $70.04 \pm 6.91$ | $83.41 \pm 6.34$ | $\mathbf{89.51 \pm 1.60}$ | $\mathbf{77.50 \pm 2.67}$ | $\mathbf{72.49 \pm 3.53}$ |
| | S | $\mathbf{74.58 \pm 6.51}$ | $\mathbf{86.67 \pm 3.83}$ | $89.30 \pm 1.35$ | $76.19 \pm 3.35$ | $72.27 \pm 5.49$ |
| SGC | – | $69.18 \pm 6.63$ | $79.79 \pm 8.91$ | $84.67 \pm 6.29$ | $76.89 \pm 4.06$ | $67.37 \pm 5.79$ |
| | P | $70.08 \pm 7.24$ | $81.02 \pm 7.98$ | $87.93 \pm 1.81$ | $78.64 \pm 2.83$ | $67.89 \pm 7.36$ |
| | S | $\mathbf{74.57 \pm 4.2}$ | $\mathbf{85.39 \pm 5.99}$ | $\mathbf{89.14 \pm 1.22}$ | $\mathbf{81.01 \pm 1.63}$ | $\mathbf{69.95 \pm 6.75}$ |
| GAT | – | $72.22 \pm 2.10$ | $82.41 \pm 1.85$ | $86.97 \pm 4.61$ | $73.40 \pm 5.16$ | $66.16 \pm 7.77$ |
| | P | $75.46 \pm 2.23$ | $85.46 \pm 1.74$ | $89.80 \pm 4.03$ | $74.14 \pm 5.17$ | $\mathbf{69.97 \pm 5.87}$ |
| | S | $\mathbf{77.61 \pm 1.33}$ | $\mathbf{85.73 \pm 2.02}$ | $\mathbf{90.02 \pm 3.4}$ | $\mathbf{75.96 \pm 5.12}$ | $67.87 \pm 5.41$ |
| GraphSAGE | – | $60.91 \pm 3.02$ | $78.05 \pm 3.59$ | $85.31 \pm 6.47$ | $71.84 \pm 7.85$ | $\mathbf{63.98 \pm 8.05}$ |
| | P | $\mathbf{63.09 \pm 2.86}$ | $77.86 \pm 5.78$ | $84.96 \pm 6.29$ | $69.23 \pm 7.57$ | $63.26 \pm 3.31$ |
| | S | $62.84 \pm 1.93$ | $\mathbf{79.61 \pm 4.62}$ | $\mathbf{86.97 \pm 5.38}$ | $\mathbf{72.25 \pm 12.93}$ | $62.77 \pm 4.34$ |

