# OpenReview forum: "GATTA: Graph Active Learning with Test-Time Augmentation"
_TMLR — Under review for TMLR_

### Review · Reviewer_NnbY · 2026-05-23

**Summary Of Contributions:**

The paper proposes an active learning strategy for selecting nodes in a graph, incrementally, for learning node labels via an active learning strategy. The acquisition function is based on an aggregation of predictions of augmented graphs, which is claimed to be novel. Experiments show gains for the proposed strategy with two datasets.

I have to say that personally I am not an expert on active learning, but the paper is easy to follow and is not heavy in hard to grasp jargon, which makes it very useful for a wide audience.

**Audience:**

Yes

**Audience Explanation:**

Active learning in general, and graph active learning in particular is a useful practical method for the cases when label acquisition is much more expensive than model training, such as human in the loop. Creating more sample efficient strategies mitigates the data labeling burden.

**Claims And Evidence:**

No

**Claims Explanation:**

1. The paper uses a constant set of hyperparameters. The improvements observed might be due to hyperparameter choice, and not due to the method itself. Even though in practice we might choose one set of hyperparameters, from a scientific perspective it's not convincing.
2. The evaluation protocol is an important aspect of the experimental section. At first I was unable to understand what exactly is being done, until I reached the appendix. Other readers might also not be convinced by experimental section without seeing the evaluation protocol.
3. It is important to clarify what kind of "graph learning" is being done here. We are not labeling graphs, but labeling nodes in one big graph (unless I've missed something?). It appears the method has not been tried in the setting of a dataset consisting of a large number of graphs. It's hard to understand what the method applies to without clarifying.

**Requested Changes:**

1. Please move the evaluation protocol to the experimental section. It's hard to grasp what exactly is being tested without it. It's not some "side information"
2. Please explain what kind of graph learning is being targeted (supervised, one graph, node labeling).
3. Explicitly explain why do you believe your claims are valid even though the gains could, in theory, be attributed to the hyperparameter choice that favors your method.

---

> ### Author Response · Authors · 2026-07-12
> **Rebuttal**
>
> Dear Reviewer,
>
>
> We thank you for the positive assessment of the paper and for highlighting its clarity and accessibility. We appreciate the constructive suggestions, all of which have helped improve the presentation of the manuscript.
>
> ## Evaluation protocol and experimental setting
>
> We agree that the evaluation protocol is central to understanding the experiments and should not have been confined to the Appendix. Following your suggestion, we moved the complete Active Learning Protocol into the main paper. This section now explicitly describes the acquisition procedure, retraining protocol, acquisition budget, and evaluation methodology.
>
> We also clarified the learning setting throughout the paper. We now explicitly state that our work considers supervised transductive node classification, where labels are acquired for nodes within a single graph, rather than graph-level classification over a collection of independent graphs. We believe this makes the scope and applicability of the proposed method considerably clearer.
>
> ## Hyperparameter choice
>
> We appreciate this comment and have expanded the discussion of hyperparameter selection. The hyperparameters specific to GATTA (augmentation type, augmentation strength, consistency filtering, and ensemble size) are not chosen arbitrarily but are derived from the sensitivity analysis presented in Section 5. Rather than optimizing these parameters separately for each dataset or acquisition strategy, we intentionally use a common configuration across experiments to evaluate the robustness of the proposed framework. Importantly, the sensitivity analysis not only motivates the selected default configuration, but also demonstrates that GATTA consistently improves performance across a broad range of hyperparameter settings.
>
> Furthermore, GATTA operates purely at inference time and does not modify the underlying GNN training procedure. Consequently, all model training hyperparameters are identical between the original acquisition strategies and their GATTA-enhanced counterparts and follow the settings used in the original implementations and prior work. This ensures that the observed improvements are attributable to the proposed uncertainty estimation procedure rather than differences in model optimization.

---

### Review · Reviewer_bcVX · 2026-06-19

**Summary Of Contributions:**

This paper introduces GATTA, a framework that incorporates test-time augmentation (TTA) into graph active learning. The key idea is to generate multiple augmented graph views during inference and aggregate predictions to obtain more reliable uncertainty estimates for node acquisition. The authors propose two aggregation variants:
* GATTA-P: aggregate predictions first and then compute acquisition scores.
* GATTA-S: compute acquisition scores on each augmented view and then aggregate scores.
To address the challenge that graph augmentations may not preserve labels, the paper introduces a consistency-based filtering mechanism that discards augmented views whose predictions disagree with the original graph prediction.

**Audience:**

Yes

**Audience Explanation:**

Same as above, this method makes some senes.

**Broader Impact Concerns:**

The work is primarily methodological and does not introduce obvious societal risks beyond those already associated with graph neural networks and active learning.

**Claims And Evidence:**

Yes

**Claims Explanation:**

(1) Novel and practically useful idea
Applying test-time augmentation to graph active learning is a natural yet underexplored direction. While TTA has been extensively studied in computer vision, its integration into graph active learning is novel and well motivated. The paper clearly positions itself relative to existing graph augmentation and graph active learning literature.

(2) Simple framework with strong empirical gains
One of the strongest aspects of the work is its simplicity. GATTA can be applied as a plug-and-play module without modifying the underlying GNN architecture or retraining procedure. The empirical results show consistent gains for Entropy and Least Confidence across datasets, often outperforming more sophisticated methods such as AGE and approaching or surpassing GEEM.

**Requested Changes:**

1. Add stronger theoretical motivation
    * Explain why TTA should improve epistemic uncertainty estimation on graphs.
    * Analyze when consistency filtering is expected to help or hurt.

---

> ### Author Response · Authors · 2026-07-12
> **Rebuttal**
>
> Dear Reviewer,
>
>
> We thank you for the positive assessment of our work and for recognizing both its practical value and the novelty of introducing test-time augmentation into graph active learning. We especially appreciate the suggestion to strengthen the conceptual motivation behind GATTA.
>
> ## Stronger theoretical motivation
> We agree that the original manuscript focused primarily on the algorithmic description and empirical evaluation, while providing only limited intuition for why test-time augmentation improves uncertainty estimation on graphs. In the revised manuscript, we introduced a theoretical motivation by interpreting GATTA as estimating the expected acquisition score over a stochastic graph augmentation distribution. Specifically, GATTA is formulated as a Monte Carlo estimator over a local neighborhood of graph perturbations, providing a principled interpretation of why aggregating predictions across augmented graph views can produce more robust uncertainty estimates.
>
> We also expanded the discussion of the proposed consistency filtering mechanism. Rather than presenting it purely as an algorithmic heuristic, we now interpret filtering as approximating a conditional expectation over model-consistent perturbations, where prediction consistency with the original graph serves as a practical proxy for local semantic consistency. This interpretation explains why filtering is expected to improve uncertainty estimation when graph augmentations occasionally introduce semantic drift, while also clarifying its limitations (e.g., when the original prediction is incorrect or disagreement across augmented views reflects genuine predictive uncertainty rather than augmentation artifacts). In the appendix, we provide an information-theoretic interpretation for entropy-based acquisition functions that offers a plausible explanation for the differing empirical behavior of GATTA-P and GATTA-S.
>
> We believe these additions provide a clearer conceptual foundation for the proposed method and strengthen the connection between the algorithmic design and the empirical observations.

---

### Review · Reviewer_oyp8 · 2026-06-29

**Summary Of Contributions:**

Summary

This paper introduces GATTA, a framework that integrates TTA into graph active learning pipelines. GATTA generates multiple augmented graph views during inference, aggregates predictions across these views to produce more reliable uncertainty estimates, and uses a consistency-based filtering mechanism to discard label-inconsistent perturbations. The paper evaluates GATTA across 5 graph datasets, 2 GNN architectures, and 6 acquisition strategies, with the central claim that simple uncertainty-based methods enhanced with TTA can match or outperform sophisticated acquisition functions at lower computational cost.

Strengths

- The consistency-based filtering mechanism reflects a genuine understanding of graph-specific challenges (structural perturbations can alter node semantics, unlike image rotations/crops).
- The four deployment guidelines (Section 6) are actionable and well-supported by the sensitivity analysis.

Weaknesses
- The core idea (TTA for active learning) has been explored in computer vision, and GATTA's methodological contributions primarily consist of domain adaptation rather than fundamental technical innovation.
- The active learning protocol uses an impractical per-round single-node acquisition with a tiny budget (4C total), and the evaluation is entirely restricted to homophilic graphs in transductive settings.
- Only 3 of 5 datasets are shown in Table 2 (with Amazon datasets omitted), and the F4 claim ("GATTA-S consistently outperforms GATTA-P") is contradicted by GCN results where GATTA-P wins on 2 of 3 reported datasets.
- No standard deviations, confidence intervals, or statistical significance tests are reported, making it impossible to verify whether observed improvements are genuine or noise.

**Audience:**

Yes

**Audience Explanation:**

Graph active learning is an important problem domain (scientific discovery, drug design, social network analysis), and the finding that simple methods can be enhanced rather than replaced has practical value for practitioners who may lack resources to implement complex acquisition functions.

**Claims And Evidence:**

No

**Claims Explanation:**

- All reported improvements are point estimates without standard deviations, confidence intervals, or p-values. In active learning (an inherently high-variance setting), improvements of +1-3% may well reflect random fluctuation. The reader has no basis to judge whether GATTA-S's +2.87% average improvement for Least Confidence is statistically distinguishable from zero.
- Table 2 presents results for only 3 citation networks (Citeseer, CoraML, PubMed), omitting Amazon Photos and Amazon Computers. While these are mentioned in text, the omission prevents direct verification of cross-domain patterns. More critically, the paper reports ESP averages -0.19% overall, but the displayed datasets show ESP performing positively (+1.22% average across displayed results), implying much stronger negative results on the omitted datasets.

**Requested Changes:**

- Discuss TAAL (Gaillochet et al., 2022) in Related Work. Explain why extending TTA+AL from images to graphs is non-trivial and what graph-specific challenges required novel solutions.
- Extend the active learning protocol to include batch acquisition (e.g., 10, 50, 100 nodes per round) and larger budgets (at least 100-200 nodes or 5-10% of dataset).
- Add experiments on heterophilic graph datasets (e.g., Chameleon, Squirrel, Actor, Cornell/Texas/Wisconsin). All 5 current datasets are homophilic.
- Provide deeper analysis of negative results: ESP decline, GEEM decline, and especially the -8.39% failure of TTA+MCD on AmazonComputers. Each deserves a plausible mechanistic hypothesis.
- Report standard deviations/confidence intervals for all results in Table 2. Add error bars to Figure 3 learning curves.

---

> ### Author Response · Authors · 2026-07-12
> **Rebuttal**
>
> Dear Reviewer,
>
> We thank you for the thorough and constructive review, and for recognizing both the practical value of the proposed framework and the graph-specific motivation behind the consistency filtering mechanism. We appreciate the detailed suggestions, which have significantly helped improve the manuscript.
>
> ## Novelty and relation to prior work
> We agree that test-time augmentation has previously been explored in computer vision. In the revised manuscript, we expanded the Related Work to discuss TAAL (Gaillochet et al., 2022) and clarify why extending test-time augmentation to graph active learning is not a straightforward transfer. Unlike image augmentations, graph perturbations directly modify the structural context and node features used by GNNs, meaning that semantic preservation is no longer guaranteed. This graph-specific challenge motivated the proposed consistency filtering mechanism, which we now interpret as approximating uncertainty estimation over model-consistent perturbations using prediction consistency as a practical proxy.
>
> More broadly, we emphasize that the contribution of GATTA extends beyond introducing a graph-specific test-time augmentation framework. To the best of our knowledge, it is the first work to systematically study how test-time augmentation should be deployed in graph active learning by analyzing its interaction with acquisition strategies, aggregation mechanisms, augmentation operators, and computational trade-offs.
>
>
> ## Experimental protocol
> Following your suggestion, we extended the evaluation to include batch acquisition experiments with acquisition sizes of 5, 10, 50, and 100 nodes per iteration, together with a labeling budget of 200. These experiments demonstrate that the proposed framework remains effective beyond the single-node acquisition protocol, while also providing additional insights into when test-time augmentation is most beneficial .Since these experiments investigate the sensitivity of GATTA to the active learning protocol rather than introduce a new benchmark comparison, we include them in the appendix.
>
>
> ## Statistical reporting
> We agree that prominently reporting uncertainty estimates is important. Standard deviations of 25 repeats for all experiments were already included in the supplementary material of the original submission. To improve their visibility, the revised manuscript now reports uncertainty alongside the learning curves, making the robustness of the reported improvements easier to assess and also explicitly reference Table 4, which includes final accuracies with variances.
>
> ## Presentation of experimental results
> We appreciate that the presentation of Table 2 could have been clearer, however all five datasets (Citeseer, CoraML, PubMed, Amazon Photos, and Amazon Computers) were included in Table 2 and contribute to all reported averages. To avoid ambiguity, we clarified this explicitly in the revised manuscript.
> We also revised Finding F4 from "GATTA-S consistently outperforms GATTA-P" to a more precise statement reflecting that GATTA-S performs better on average and in most experimental settings, while GATTA-P can outperform it for particular dataset/model combinations.
>
> ## Analysis of negative results
> Following your suggestion, we expanded the discussion of the observed negative results. Rather than analyzing each acquisition strategy independently, we now provide a unified interpretation based on the role of GATTA as an uncertainty refinement module. Specifically, we argue that GATTA primarily improves the predictive distribution supplied to the acquisition function rather than the acquisition objective itself. This naturally explains why simple uncertainty-based methods (e.g., Entropy and Least Confidence) benefit the most, while acquisition strategies that already incorporate sophisticated uncertainty estimation or expected error reduction (e.g., ESP, GEEM, and MC Dropout) exhibit smaller or less consistent improvements. We also discuss why prediction aggregation (GATTA-P) is substantially more sensitive to inconsistent perturbations than score aggregation (GATTA-S), providing a principled interpretation of the observed empirical behavior.
>
> ## Heterophilic graphs
> We agree that evaluating heterophilic graph benchmarks is an important direction for future work. Our goal in this paper, however, is to study test-time augmentation within the standard transductive graph active learning protocol adopted by prior work. Extending the evaluation to heterophilic benchmarks would likely require reconsidering both the underlying GNN architectures and suitable graph augmentation strategies, making it a substantially broader investigation than the scope of the present work. We therefore discuss this limitation explicitly in the revised manuscript and identify evaluation on heterophilic graphs as an important direction for future research.

---

### Author Response · Authors · 2026-07-12
**Rebuttal**

We thank all reviewers for their constructive feedback. Following their suggestions, we revised both the methodology and experimental evaluation. We uploaded a revision of the manuscript, where we highlighted the updated sections with red. The main changes are summarized below.
 - Strengthened theoretical motivation. We now formulate GATTA as Monte Carlo estimation over a stochastic graph augmentation distribution, interpret consistency filtering as approximating a conditional expectation over model-consistent perturbations, and provide an information-theoretic interpretation for entropy-based acquisition functions that explains the differing behavior of GATTA-P and GATTA-S.

 - Expanded experimental evaluation. We added experiments with larger acquisition batch sizes (5, 10, 50, and 100 nodes per iteration) and extended labeling budgets (200 acquired labels), demonstrating that GATTA remains effective beyond the standard single-node acquisition protocol. These experiments are now also included in the appendix.
 - Improved experimental presentation. We moved the Active Learning Protocol from the appendix to the main paper, clarified the transductive node classification setting, improved the presentation of statistical uncertainty by explicitly reporting variances alongside learning curves and referencing the complete statistical tables.
 - Expanded analysis and discussion. We extended the discussion of the observed empirical behavior, providing a unified interpretation of why GATTA benefits simple uncertainty-based acquisition strategies while exhibiting smaller or less consistent gains for more sophisticated methods. We also refined the discussion of the differing robustness of GATTA-P and GATTA-S.

- Clarified scope and limitations. We expanded the discussion of the applicability of the proposed framework, explicitly discussing the current focus on homophilic transductive node classification and identifying heterophilic graphs as an important direction for future work.